# Frozen-DETR: Enhancing DETR with Image Understanding from Frozen Foundation Models

**Shenghao Fu**[1,3,4], **Junkai Yan**[1,3,4], **Qize Yang**[3†], **Xihan Wei**[3],
**Xiaohua Xie**[1,4,5*] **Wei-Shi Zheng**[1,2,4,6*]

[1]School of Computer Science and Engineering, Sun Yat-sen University, China;
[2]Peng Cheng Laboratory, Shenzhen, 518055, China;
[3]Tongyi Lab, Alibaba Group;
[4]Key Laboratory of Machine Intelligence and Advanced Computing, Ministry of Education, China;
[5]Guangdong Province Key Laboratory of Information Security Technology, China;
[6]Pazhou Laboratory (Huangpu), Guangzhou, Guangdong 510555, China
{fushh7,yanjk3}@mail2.sysu.edu.cn, xiexiaoh6@mail.sysu.edu.cn, wszheng@ieee.org
https://github.com/iSEE-Laboratory/Frozen-DETR

## Abstract

Recent vision foundation models can extract universal representations and show impressive abilities in various tasks. However, their application on object detection is largely overlooked, especially without fine-tuning them. In this work, we show that frozen foundation models can be a versatile feature enhancer, even though they are not pre-trained for object detection. Specifically, we explore directly transferring the high-level image understanding of foundation models to detectors in the following two ways. First, the class token in foundation models provides an in-depth understanding of the complex scene, which facilitates decoding object queries in the detector's decoder by providing a compact context. Additionally, the patch tokens in foundation models can enrich the features in the detector's encoder by providing semantic details. Utilizing frozen foundation models as plug-and-play modules rather than the commonly used backbone can significantly enhance the detector's performance while preventing the problems caused by the architecture discrepancy between the detector's backbone and the foundation model. With such a novel paradigm, we boost the SOTA query-based detector DINO from 49.0% AP to 51.9% AP (+2.9% AP) and further to 53.8% AP (+4.8% AP) by integrating one or two foundation models respectively, on the COCO validation set after training for 12 epochs with R50 as the detector's backbone.

## 1 Introduction

Understanding an image at both global and local levels is a key factor for a wide range of vision perception tasks. Typically, in object detection, an in-depth understanding of the image can assist model reasoning under many challenging situations. First, the conflict between detecting the whole object and parts of it always exists in object detection since parts of the object are also annotated in many scenarios, *e.g.*, objects under occlusion. With a global understanding of the image, detectors can recognize different parts of the same object and detect the objects as completely as possible, as shown in Figure 1 (a). Second, the co-occurrence of objects can facilitate finding some missing objects. In Figure 1 (b), a man is sitting on something, from which we can infer that it is a bench with a strange appearance. And the bottle aside the man may help us find the bottle in his hand. Moreover,

---

*: Corresponding authors are Xiaohua Xie and Wei-Shi Zheng. †: Project Lead. This work was done when Shenghao Fu and Junkai Yan were interns at Alibaba.

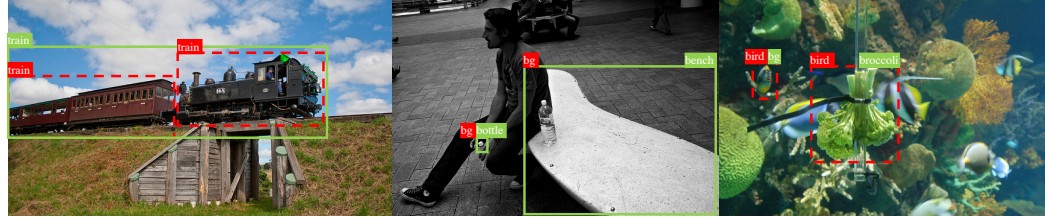

(a) clarify the relation between parts and the whole  (b) find missing objects  (c) correct wrong classification

Figure 1: An in-depth understanding of the image provides useful information for detecting objects. (a) With the rich context, the relation between object parts and the whole object can be clarified. (b) Some objects with severe occlusion or unusual appearance can be discovered by co-occurrence or interaction with other objects. (c) And similar objects can be distinguished by some salient features. The red and green boxes represent incorrect and correct predictions, respectively.

some salient features can also help us distinguish similar objects, *i.e.*, distinguishing the fish and broccoli from birds as shown in Figure 1 (c). Prior efforts to enhance detectors' image understanding ability include using a strong backbone with a large receptive field [30, 23, 18] and explicitly injecting scene information to detectors [48, 14, 12]. Recent query-based detectors [3, 73, 22, 66] modeling object queries via the global self-attention mechanism also enjoy global reasoning. In this work, we explore the image understanding ability from outside knowledge rather than the detector itself.

In light of this, we focus on the recently attractive foundation models, which have shown an impressive understanding of the image via large-scale pre-training, even without task-specific data. For example, DNF CLIP [21] with ViT-H can achieve 84.4% zero-shot classification accuracy on ImageNet [17], achieving similar results to the same ViT-H [19] (85.1%) trained in a supervised way. Benefiting from the advanced architecture, extensively collected data, and well-designed pre-training tasks, the off-the-shelf foundation models are already equipped with strong image understanding abilities.

In this work, we propose Frozen-DETR, which uses a frozen foundation model as a plug-and-play module to boost the performance of query-based detectors. Instead of using the foundation model as a backbone, we regard it as a feature enhancer from two perspectives: First, to utilize the global image understanding ability of foundation models, we take the class token from them as the full image representation, termed image query. The image query is concatenated with object queries and facilitates decoding object queries in the decoder by providing a complex scene context. Second, the fine-grained patch tokens with high-level semantic cues from foundation models are considered as another level of the feature pyramid, which is then fused with the detector's feature pyramid via the encoder layers. The foundation model is parallel with the backbone and frozen during training.

Compared with methods that use the foundation model as a learnable backbone [40, 13] or a frozen backbone [58, 44], our method enjoys the advantages in the following three aspects: **a) No architecture constraint**. Since we do not require the foundation model to extract multi-scale features, any architecture, CNNs, ViTs, or hybrid ones, can be used as the foundation model's architecture. Besides, the detector and the foundation model can use different structures. **b) Plug-and-play**. Our method can be plugged into various query-based detectors without modifying the detector's structure, the foundation model's structure, and the training recipe. **c) Asymmetric input size**. We use the foundation model as a feature enhancer rather than a backbone. The input image size of the foundation model can be much smaller than the one for the backbone (*e.g.*, 336 *vs.* 1333). Considering the asymmetric input size, we can use a large foundation model with an acceptable computation burden.

We find that CLIP [54] is one of the best candidates for Frozen-DETR, and its high-level semantic understanding significantly enhances the classification ability of DETRs in many challenging scenarios. Taking the well-known DINO [66] detector as the baseline, we boost its performance to 53.2% AP (+2.8%) on the COCO dataset. On the large vocabulary LVIS dataset, Frozen-DETR increases an impressive 6.6% AP. Considering the long-tail data distribution, Frozen-DETR increases 8.7% APr and 7.7% APc, showing the potential to alleviate the class imbalance problem. On the open-vocabulary scenario, we increase by 8.8% novel AP, showing strong open-vocabulary ability. Further, Frozen-DETR achieves almost the same performance on the COCO-O [50] dataset compared with the one on the in-domain COCO dataset, showing great domain generalization ability.

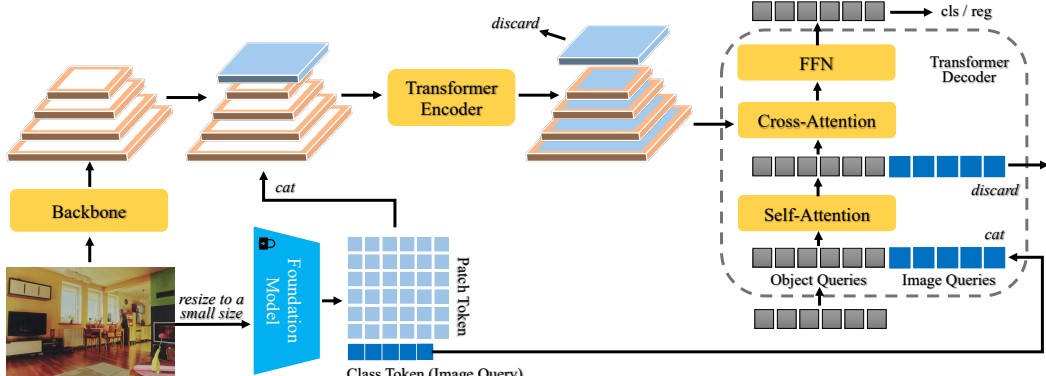

Figure 2: The overview of Frozen-DETR. Instead of serving as a backbone, we exploit the frozen foundation model from the following two aspects: First, the patch tokens are reshaped to a 2D feature map and are concatenated with feature maps from the backbone before the encoder. After feature fusion, the patch tokens are discarded. Second, the image query representing the whole image, *i.e.*, the class token from the foundation model, interacts with object queries in the self-attention layer of each decoding stage. Using the frozen foundation model as a feature enhancer makes the detector inherit the strong ability to understand high-level semantics.

## 2  Related Works

**Foundation models and their application in object detection.** Recent vision foundation models can be divided into supervised ones [57, 54, 39, 35] and self-supervised ones [11, 4, 60, 27, 1, 61, 10, 70, 52]. Training foundation models in a supervised manner need to collect web-scale high-quality datasets, such as image classification dataset ImageNet-22k [17] (DEiT-III [57]), image-text pair dataset WIT [54] (CLIP), grounding dataset GoldG [39] (GLIP), and segmentation dataset SA-1B [35] (SAM). Training with gold annotations from humans, these models can be directly applied to many downstream tasks without fine-tuning. However, the human-annotated datasets are hard to scale up due to intensive labor. Self-supervised learning is an alternative. With well-designed pre-tasks, *e.g.*, contrastive learning [11, 4, 60, 65], mask image modeling [27, 1, 61, 10], or their combination [70, 52], models can learn distinctive representations without human annotations. But to unleash the power of self-supervised learning on downstream tasks, models should be fine-tuned with task-specific data.

Pre-training-then-fine-tuning is a common paradigm to use the pre-trained foundation models in object detection. By transferring the pre-trained knowledge, the detector can converge much faster than training from scratch [28], shows better advantages in robustness and uncertainty [31], and even gains higher performance. However, in this paradigm, the detector should use the same backbone as the pre-trained foundation model to transfer the pre-trained weights. Unfortunately, not all backbones are suitable for dense prediction tasks. Thus, many task-oriented designs are introduced to compensate for the structure inadaptability, *e.g.*, ViT-Adapter [13]. Besides, fine-tuning a large foundation model is not always acceptable due to resource constraints. To this end, a few works [58, 44] explore using frozen foundation models as the backbone. To make the frozen backbone work in object detection, modifying the structure (heavy neck and head) and the training recipe (long training schedule) is necessary. In this work, we explore using the foundation model as a feature enhancer rather than the backbone, which can avoid the problems mentioned above.

**Query-based object detectors.** Different from traditional detectors [55, 29, 42, 51], DETR [3] formulates object detection as a set prediction task, which is an end-to-end model without using non-maximum suppression (NMS). The following improvements mainly include advanced formulations of object queries [73, 24, 62, 45, 47], stabilizing bipartite matching [36, 66, 46], providing more supervision signals [8, 6, 34, 74, 68], alleviating conflict or competition between object queries [67, 32, 53] and knowledge distillation [5, 33, 9]. In this work, we explore enhancing DETR from another aspect by utilizing the image understanding ability of off-the-shelf foundation models.

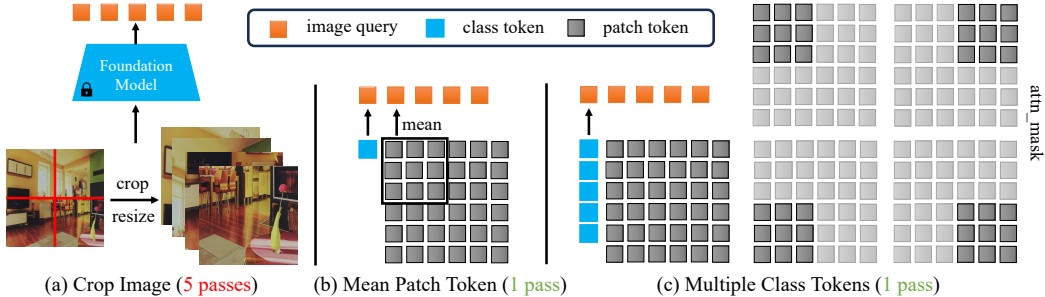

(a) Crop Image (5 passes)     (b) Mean Patch Token (1 pass)     (c) Multiple Class Tokens (1 pass)

Figure 3: Different implementations to extract image queries for sub-images. (a) Forwarding each sub-image individually to the model and selecting the class token as the image query. (b) Using the mean features of the patch tokens as the image queries for sub-images. (c) Using the replicated class tokens as the image queries for sub-images but these class tokens are constrained by attention masks.

## 3 Foundation Models as Feature Enhancers

### 3.1 Preliminaries

**Vision Transformer (ViT).** Recently, transformer [59] has been shown to be a scalable architecture [16, 2, 37] and many foundation models [54, 52, 61, 27, 57] are based on it. Different from CNNs, ViT [19] first splits images into patches and projects each patch to a patch token. These tokens are flattened in spatial dimensions and modeled in a sequence manner. In addition to patch tokens, a learnable class token is prepended to the patch sequence. The patch tokens preserve the local details for each patch, while the class token represents the global information for the whole image. We take both the class token and patch tokens into account to enhance the detector's ability.

**DETR.** A common DETR-like [3] detector includes 3 parts: backbone, encoder, and decoder. The backbone can be any architecture that produces feature pyramids. The encoder uses deformable attention [73] to enhance the feature maps while avoiding unaffordable computation costs from global self-attention. Taking the refined feature maps and some object queries as inputs, the decoder aims to refine the object queries layer by layer and predicts the class label and box coordinates for each object query. In the subsequent improvement, each object query can be divided into two parts: content vector representing the feature for each object, and position vector indicating the location for each object. During the training, each object query will be optimized towards a single object or background.

In this work, we employ patch tokens from foundation models to enhance the DETR's encoder via feature fusion and facilitate the decoding process of DETR's decoder with the class token from foundation models.

### 3.2 Enhancing Decoder by Treating Class Tokens as Image Queries

We introduce a new kind of query into the decoder, termed image query. Different from conventional object queries whose content vector represents a single object and the position vector is the bounding box of the object, the image query represents the whole image, and its box is the full image boundary. Since pre-trained foundation models have a strong ability to understand complex images at a global level, we take advantage of their scene-understanding ability and treat the class token as the image query. With the image query as context, object queries can be better classified.

Specifically, as shown in Figure 2, for each image, we extract the image query by passing the raw image to the frozen foundation model. The image is resized to the pre-training image size of the foundation model. Then, in each decoder layer, we project the image query to the same dimensions as object queries and concatenate these two kinds of queries before feeding them into the self-attention module. In the following self-attention computation, object queries can interact adaptively with the image query to absorb high-level image understanding from the foundation model. Finally, the image query is discarded after passing the self-attention module.

Since a single global image query may overlook several inconspicuous small objects, to compensate for this and provide fine-grained local context, we propose to split the image into multiple sub-images evenly and equip each sub-image with an image query. For example, if we split the image into

Table 1: Effect of image queries with different detector backbone.

| Detector Backbone | Image Query | AP | $AP_{50}$ | $AP_{75}$ | $AP_s$ | $AP_m$ | $AP_l$ |
|---|---|---|---|---|---|---|---|
| IN-1k R50 | | 44.1 | 63.3 | 47.5 | 26.7 | 47.0 | 60.3 |
| IN-1k R50 | ✓ | 45.0 | 64.8 | 48.7 | 27.5 | 47.3 | 62.1 |
| CLIP R50 | | 45.5 | 64.9 | 49.1 | 28.7 | 48.8 | 59.1 |
| CLIP R50 | ✓ | 46.1 | 66.2 | 49.7 | 28.8 | 48.9 | 61.2 |

$2 \times 2 = 4$ sub-images, we will obtain 5 image queries, including an original global image query and four local image queries. A straightforward method to obtain multiple image queries is to crop the sub-images from the original one and pass them to the foundation model individually, as shown in Figure 3 (a). However, forwarding the foundation model multiple times is always time-consuming, especially when the foundation model is large enough.

We provide two fast implementations to tackle the above limitation. In the first implementation (Figure 3 (b)), we spatially split the entire patch tokens into multiple groups of tokens corresponding to the sub-images, where the mean feature of each token group is regarded as its local image query. The second one employs extra replicated class tokens to extract local image queries. We apply a masked attention operation to ensure each local class token focuses on its corresponding sub-image by restricting the interaction to a local area, as shown in Figure 3 (c). These two fast implementations can obtain global and all local image queries in a single forwarding process, thus significantly reducing the computation costs. Finally, all image queries are concatenated with object queries.

### 3.3 Enhancing Encoder by Feature Fusion

In addition to utilizing the global scene understanding of foundation models in the decoder, we further reuse the patch tokens with fine-grained semantic details. Specifically, as shown in Figure 2, we take patch tokens from the last layer of foundation models and reshape them to a 2D feature map. This 2D feature map is then concatenated with the feature maps from the detector's backbone and passed to the encoder, allowing adaptive fusion. After fusion, the backbone features are expected to assimilate the high-level semantic understanding in foundation models. We empirically find that we can simply discard the patch tokens after feature fusion, and using the enhanced feature maps from the detector's backbone is sufficient. Note that the feature map of patch tokens ($24 \times 24$ for an image with input size 336 and patch size 14) is much smaller than feature maps from the backbone ($167 \times 167$ for an image with input size 1333 and stride 8), the additional computation burden is acceptable.

## 4 Experiments

In the following subsections, we first explore the best practices of using foundation models as plug-and-play modules and then show the highly engaging performance under various scenarios. Since the detector DINO [66] and the self-supervised foundation model DINOv2 [52] share the same name and may cause confusion. We rename the detector DINO as **DINO-det** in the following.

### 4.1 Image Queries Provide Complex Contexts

**Setting.** In this subsection, we take AdaMixer [24] as the baseline to explore the usage of image queries since it is a decoder-only detector and converges fast with a small computation cost. All experiments are conducted on the COCO [43] dataset with 300 queries, 12 training epochs, and 4 V100 GPUs. Unless otherwise specified, we employ the ImageNet-1k [17] supervised pre-training ResNet-50 (R50) [30] as the backbone of the detector.

**Are image queries helpful?** In this ablation study, we use the OpenAI CLIP ViT-L-14-336 [54] to extract the global image query. As shown in the upper part of Table 1, with only one global image query appended to object queries, the AP significantly increases by 0.9%. Although the global image query represents the information from a global view, the AP for both small (+0.8% $AP_s$) and large (+1.8% $AP_l$) objects are increased.

**Are image queries equal to a better detector backbone?** We change the detector's backbone to the CLIP R50 and train it along with the detector. As shown in the lower part of Table 1, using a stronger backbone improves the AP to 45.5%, which is in line with common experience. Nevertheless, using

Table 2: Effect of using different foundation models to extract image queries.

| Foundation Model | AP | $AP_{50}$ | $AP_{75}$ | $AP_s$ | $AP_m$ | $AP_l$ |
|---|---|---|---|---|---|---|
| N/A | 44.1 | 63.3 | 47.5 | 26.7 | 47.0 | 60.3 |
| Self | 44.2 | 63.3 | 47.8 | 26.4 | 47.4 | 60.1 |
| DEiT-III [57] | 45.1 | 64.5 | 48.5 | 26.7 | 47.6 | 62.1 |
| OpenAI CLIP [54] | 45.0 | 64.8 | 48.7 | 27.5 | 47.3 | 62.1 |
| DINOv2 [52] | 44.6 | 63.9 | 48.2 | 27.6 | 47.2 | 60.6 |
| MAE [27] | 44.4 | 63.6 | 47.7 | 26.8 | 47.4 | 60.3 |
| BEiT-3 [61] | 44.1 | 63.2 | 47.9 | 26.7 | 47.0 | 60.8 |

Table 3: Effect of different implementations to extract image queries.

| Method | # Queries | AP | $AP_{50}$ | $AP_{75}$ | $AP_s$ | $AP_m$ | $AP_l$ | $t_{train}$ | $t_{infer}$ |
|---|---|---|---|---|---|---|---|---|---|
| Crop Image | 1 | 45.0 | 64.8 | 48.7 | 27.5 | 47.3 | 62.1 | 26h | 9.3fps |
| Crop Image | 1+4 | 45.7 | 65.7 | 49.5 | 28.5 | 48.8 | 62.4 | 44h | 3.7fps |
| Mean Patch Token | 1+4 | 45.3 | 65.2 | 49.2 | 27.7 | 48.2 | 61.3 | 26h | 9.1fps |
| Multiple Class Tokens | 1+4 | 45.8 | 65.7 | 49.4 | 28.7 | 48.1 | 63.0 | 26h | 9.2fps |

Table 4: Ablation studies on feature fusion in the encoder.

| Exp. | Method | AP | $AP_{50}$ | $AP_{75}$ | $AP_s$ | $AP_m$ | $AP_l$ | Mem | GFLOPs | FPS |
|---|---|---|---|---|---|---|---|---|---|---|
| 1 | baseline (no foundation model) | 49.0 | 66.5 | 53.6 | 30.6 | 52.5 | 64.2 | 13G | 279 | 9.7 |
| 2 | Exp. 1 + 5 image queries | 50.8 | 68.8 | 55.6 | 33.0 | 53.9 | 67.4 | 14G | 392 | 6.7 |
| 3 | Exp. 2 + patch tokens to encoder | 52.6 | 70.9 | 57.4 | 35.0 | 56.1 | 70.4 | 15G | 400 | 6.5 |
| 4 | Exp. 3 + patch tokens to decoder | 52.7 | 71.1 | 57.6 | 34.5 | 56.0 | 70.8 | 15G | 400 | 6.5 |

an additional global image query further increases 0.6% AP, showing that using the foundation model as an image query is orthogonal to using the foundation model as a backbone.

**Which foundation model is more suitable to extract image queries?** In this experiment, we choose 5 representative foundation models with different pre-training methods: DEiT-III [57] (ImageNet-22k supervised pre-training), CLIP [54] (image-text pair alignment), MAE [27], BEiT-3 [61] (masked data modeling) and DINOv2 [52] (masked data modeling and online self-distillation). All the models use the ViT-L and the input image sizes are adjusted to 336 by interpolating the positional embedding. For models that do not use the class token during the pre-training (MAE and BEiT-3), we use the mean patch tokens as the image query. Besides, we also use the mean feature from the detector's backbone as the image query for a clear comparison, denoted as "Self". As shown in Table 2, using the mean backbone feature of the detector as the image query does not help much since the DETRs already model object queries with global self-attentions. Besides, we find that the foundation models which are pre-trained using human labels (DEiT-III and CLIP) perform better than the self-supervised counterparts, perhaps the self-supervised ones lack the high semantic from human supervision thus the features can not be utilized in the down-stream tasks without fine-tuning. Another possible reason is that models pre-trained with masked data modeling could focus more on local texture details. Since image-text pairs are more scalable than image classification datasets, we choose the OpenAI CLIP (ViT-L-14-336) as the foundation model in the following experiments.

**Extracting image queries with fast implementations.** As demonstrated in Section 3.2, using some additional local image queries to represent sub-images can preserve more details for the local context. Table 3 illustrates the results of different implementations. The training time $t_{train}$ is the total time for training 12 epochs and the inference time is tested on a single V100 GPU with batch size 1. As shown in Table 3, cropping sub-images from the original one can increase the AP to 45.7% but with more training time (+18h) and inference time (+150%) since images should pass through the whole foundation model for multiple times. In contrast, using the mean patch token or multiple class tokens can save much time as only one pass is needed. However, using the mean patch token is less effective since the patch token is less representative than the class token in ViT CLIP [64]. Thus, in the following experiments, we use multiple class tokens to extract multiple image queries by default.

**How many image queries do we need?** We conduct experiments with 0, 1, 5 (1+4), and 14 (1+4+9) image queries and achieve 44.1% AP, 45.0% AP, 45.8% AP, and 45.6% AP, respectively. We empirically find that using 5 image queries is enough and more image queries do not help. Thus we use 5 image queries by default.

Table 5: Ablation studies on the input image size of the foundation model.

| input image size | AP | $AP_{50}$ | $AP_{75}$ | $AP_s$ | $AP_m$ | $AP_l$ | GFLOPs | FPS |
|---|---|---|---|---|---|---|---|---|
| 224 | 51.5 | 69.7 | 55.9 | 32.5 | 54.6 | 69.7 | 333 | 7.7 |
| 336 | 52.6 | 70.9 | 57.4 | 35.0 | 56.1 | 70.4 | 400 | 6.5 |
| 448 | 53.1 | 71.6 | 58.2 | 33.9 | 56.6 | 71.0 | 494 | 4.9 |
| 560 | 52.8 | 71.1 | 57.5 | 33.4 | 56.6 | 70.5 | 615 | 3.7 |

Table 6: Ablation studies on the model size of the foundation model.

| model size | AP | $AP_{50}$ | $AP_{75}$ | $AP_s$ | $AP_m$ | $AP_l$ | GFLOPs | FPS |
|---|---|---|---|---|---|---|---|---|
| - | 49.0 | 66.5 | 53.6 | 30.6 | 52.5 | 64.2 | 279 | 9.7 |
| R101 (640) | 50.1 | 68.0 | 54.9 | 33.0 | 53.5 | 65.7 | 356 | 7.6 |
| ViT-B-16 (320) | 50.7 | 68.5 | 55.5 | 32.6 | 53.6 | 67.8 | 304 | 8.4 |
| ViT-L-14 (336) | 52.6 | 70.9 | 57.4 | 35.0 | 56.1 | 70.4 | 400 | 6.5 |

Table 7: Ablation studies on whether fine-tuning the foundation model (CLIP R101).

| Foundation Model | AP | $AP_{50}$ | $AP_{75}$ | $AP_s$ | $AP_m$ | $AP_l$ | Mem |
|---|---|---|---|---|---|---|---|
| N/A | 49.0 | 66.5 | 53.6 | 30.6 | 52.5 | 64.2 | 13G |
| Trainable | 49.0 | 66.7 | 53.5 | 31.1 | 52.3 | 64.3 | 17G |
| Frozen | 50.1 | 68.0 | 54.9 | 33.0 | 53.5 | 65.7 | 14G |

## 4.2 Patch Tokens Enhance Feature Fusion in Encoder

**Setting.** In this subsection, we change the baseline to Co-DINO [74], which has both transformer encoder and decoder. For fast implementation, we use four-scale feature maps (1/8, 1/16, 1/32, 1/64) and do not use co-heads. All experiments are conducted on the COCO [43] dataset with R50, 900 queries, 12 training epochs, and 4 V100 GPUs.

**Ablation studies on each component of feature fusion.** As shown in Table 4, adding image queries to the new detector also gets 1.8% AP gains (Exp. 2), showing its generalizability. If we regard the patch tokens as an additional $24 \times 24$ feature map and append it to the encoder (5-scale feature maps now, Exp. 3), the AP further increases by 1.8% AP. Since the added feature map is small and the foundation model is frozen, the additional computation cost and training time GPU memory (batch size 2 per GPU) are acceptable. We further find sending the patch tokens to the cross-attention in the decoder (Exp. 4) is unnecessary and we simply discard the patch tokens after the encoder.

**Ablation studies on the input image size of the foundation model.** In Table 5, we change the input size to 224, 336, 448, and 560, and achieve 51.5% AP (333 GFLOPs), 52.6% AP (400 GFLOPs), 53.1% AP (494 GFLOPs), and 52.8% AP (615 GFLOPs). We find that the input image size is not necessarily the larger, the better. On the one hand, the image size should not be too large since the foundation model is pre-trained under a small resolution. On the other hand, the foundation model provides a high-level semantic image understanding rather than location texture details. Thus, the large input size is not necessary. Further, the large input size brings a huge computation burden since the self-attention module in the foundation models has a quadratic complexity in the length of patch tokens. By default, we use 336 as the input image size.

**Ablation studies on the model size of the foundation model.** In Table 6, we change the foundation model with various model sizes (R101, ViT-B-16, ViT-L-14) but keep the number of patch tokens to a similar size. There is a clear trend that a stronger foundation model can achieve higher performance, showing the great scalability of our method.

**Fine-tuning or freezing the foundation model.** To make the fine-tuning affordable, we use CLIP R101 as the foundation model in this experiment. As shown in Table 7, tuning the foundation model with the detector underperforms the frozen one. We assume that training with much fewer downstream task data breaks the pre-trained representations in foundation models.

## 4.3 Main Results on COCO dataset

In this subsection, we apply our Frozen-DETR (CLIP ViT-L-14-336) to various well-known detectors, including DAB-DETR [45], DN-DETR [36], MS-DETR [68], HPR [69], DINO-det [66] and Co-DINO [74]. We strictly follow their experiment settings without changing any hyper-parameters.

Table 8: Comparisons with other query-based detectors on COCO `minival` set. *: the input size of the foundation model is 448. †: The single-scale detector uses standard attention in the encoder while Frozen-DETR uses deformable attention to fuse multi-scale features.

| Detector | Backbone | # Epochs | AP | $AP_{50}$ | $AP_{75}$ | $AP_s$ | $AP_m$ | $AP_l$ | GFLOPs | FPS |
|---|---|---|---|---|---|---|---|---|---|---|
| DETR [3] | R50 | 500 | 43.3 | 63.1 | 45.9 | 22.5 | 47.3 | 61.1 | 86 | 27.8 |
| Deformable DETR [73] | R50 | 50 | 43.8 | 62.6 | 47.7 | 26.4 | 47.1 | 58.0 | 173 | 13.4 |
| Sparse R-CNN [56] | R50 | 36 | 45.0 | 63.4 | 48.2 | 26.9 | 47.2 | 59.5 | 174 | 17.8 |
| AdaMixer [24] | R50 | 36 | 47.0 | 66.0 | 51.1 | 30.1 | 50.2 | 61.8 | 132 | 16.6 |
| DDQ DETR 4scale [67] | R50 | 24 | 52.0 | 69.5 | 57.2 | 35.2 | 54.9 | 65.9 | 249 | 8.6 |
| Group DETR (DINO-det-4scale) [8] | R50 | 36 | 51.3 | - | - | 34.7 | 54.5 | 65.3 | 279 | 9.7 |
| H-Deformable-DETR [34] | R50 | 36 | 50.0 | 68.3 | 54.4 | 32.9 | 52.7 | 65.3 | 268 | 11.0 |
| DAC-DETR [32] | R50 | 24 | 51.2 | 68.9 | 56.0 | 34.0 | 54.6 | 65.4 | 279 | 9.7 |
| DAB-DETR-DC5 [45]† | R50 | 12 | 38.0 | 60.3 | 39.8 | 19.2 | 40.9 | 55.4 | 220 | 10.2 |
| **Frozen-DETR** (DAB-DETR-DC5) | R50 | 12 | 42.0 | 63.2 | 44.9 | 22.4 | 45.4 | 61.1 | 372 | 8.5 |
| DN-DETR-DC5 [36]† | R50 | 12 | 41.7 | 61.4 | 44.1 | 21.2 | 45.0 | 60.2 | 220 | 10.2 |
| **Frozen-DETR** (DN-DETR-DC5) | R50 | 12 | 44.4 | 64.8 | 47.7 | 23.8 | 47.7 | 64.6 | 372 | 8.5 |
| DINO-det-4scale [66] | R50 | 12 | 49.0 | 66.6 | 53.5 | 32.0 | 52.3 | 63.0 | 279 | 9.7 |
| DINO-det-4scale [66] | R50 | 24 | 50.4 | 68.3 | 54.8 | 33.3 | 53.7 | 64.8 | 279 | 9.7 |
| DINO-det-5scale [66] | R50 | 24 | 51.3 | 69.1 | 56.0 | 34.5 | 54.2 | 65.8 | 860 | 4.4 |
| **Frozen-DETR** (DINO-det-4scale) | R50 | 12 | 51.9 | 70.4 | 56.7 | 33.8 | 54.9 | 69.3 | 400 | 6.5 |
| **Frozen-DETR** (DINO-det-4scale) | R50 | 24 | 53.2 | 71.8 | 58.0 | 35.1 | 56.5 | 70.6 | 400 | 6.5 |
| MS-DETR [68] | R50 | 12 | 50.0 | 67.3 | 54.4 | 31.6 | 53.2 | 64.0 | 252 | 10.8 |
| **Frozen-DETR** (MS-DETR) | R50 | 12 | 53.0 | 71.5 | 57.8 | 35.1 | 55.8 | 70.8 | 423 | 6.9 |
| DDQ with HPR [69] | R50 | 12 | 52.4 | 69.9 | 57.5 | 35.9 | 55.5 | 66.7 | 283 | 6.5 |
| **Frozen-DETR** (DDQ with HPR) | R50 | 12 | 55.7 | 73.9 | 61.3 | 38.4 | 58.8 | 72.3 | 467 | 5.2 |
| Co-DINO-5scale [74] | R50 | 12 | 52.1 | 69.4 | 57.1 | 35.4 | 55.4 | 65.8 | 860 | 4.4 |
| Co-DINO-4scale [74] | Swin-B(22k) | 12 | 56.8 | 74.9 | 62.5 | 41.7 | 60.9 | 72.8 | 513 | 6.2 |
| **Frozen-DETR** (Co-DINO-4scale) | R50 | 12 | 54.0 | 72.4 | 59.1 | 36.0 | 58.0 | 71.5 | 400 | 6.5 |
| **Frozen-DETR** (Co-DINO-4scale) | R50 | 24 | 54.3 | 72.9 | 59.2 | 36.6 | 58.0 | 72.1 | 400 | 6.5 |
| **Frozen-DETR** (Co-DINO-4scale)* | Swin-B(22k) | 12 | 57.6 | 75.8 | 63.2 | 41.4 | 61.7 | 74.3 | 732 | 3.8 |

Table 9: Results on LVIS v1 training with full annotations. *: Our implementation.

| Method | # Epochs | Backbone | AP | $AP_r$ | $AP_c$ | $AP_f$ |
|---|---|---|---|---|---|---|
| Deformable-DETR [34] | 24 | R50 | 32.2 | 20.9 | 31.1 | 38.4 |
| Detic (Deformable-DETR) [71] | 96 | R50 | 32.5 | 26.2 | 31.3 | 36.6 |
| H-Deformable-DETR [34] | 24 | R50 | 33.6 | 22.2 | 32.4 | 39.9 |
| DINO-det-4scale* | 24 | R50 | 34.4 | 22.5 | 33.4 | 40.8 |
| **Frozen-DETR** (Ours) | 24 | R50 | **41.0** | **31.2** | **41.1** | **45.1** |

As shown in Table 8, Frozen-DETR outperforms baselines ranging from 2.7% AP to 4.0% AP. The results on different detectors show the generalization ability. Although the additional patch tokens from foundation models make our Frozen-DETR also have 5 scale feature maps in the encoder, the added feature map ($24 \times 24$) is much smaller than the $C_2$ feature map in DINO-det-5scale ($333 \times 333$ for input size 1333 and stride 4). Thus the additional computation costs of Frozen-DETR ($279 \rightarrow 400$ GFLOPs) are also much smaller than DINO-det-5scale ($279 \rightarrow 860$ GFLOPs). Further, Frozen-DETR (DINO-det-4scale) and Frozen-DETR (Co-DINO-4scale) also outperform DINO-det-5scale and Co-DINO-5scale by 1.9% AP. Moreover, we can also increase 0.8% AP when using a strong backbone Swin-B [49] pre-trained on ImageNet-22k, demonstrating our great scalability. We find that large objects enjoy the most significant improvement from the image understanding of the foundation model. For example, there is an improvement of 6.3% $AP_l$ on the 12 epochs setting over DINO-det-4scale. We hypothesize that it is because large objects may easily be confused by the relation between the parts and the whole objects, as illustrated in Figure 1 (a). We also find that performance gains from more epochs for Frozen-DETR on Co-DINO (+0.3% AP) are less than Frozen-DETR on DINO-det (+1.3% AP). This is because most SOTA query-based detectors can converge extremely fast within 12 epochs. Some training strategies are excepted to further improve the performance with longer training schedules, e.g., large-scale jitter and other data augmentation.

## 4.4 Dose Frozen-DETR Work under Large Vocabulary Settings?

**Closed-set Setting.** Since Frozen-DETR has a strong advantage in classification compared with baselines benefiting from the image understanding of foundation models, we further validate the effectiveness of Frozen-DETR on the challenging LVIS v1 [26] dataset. LVIS dataset is a large vocabulary dataset (1203 classes) with long tail distribution. The classes are divided into rare, common, and frequent classes based on the number of annotations. We choose DINO-det-4scale [66] as the baseline and train the model for 24 epochs without using mask annotations. Following common practices, we use repeat factor sampling and Federated Loss [72]. As shown in Table 9, Frozen-DETR

Table 10: Results on open-vocabulary LVIS. *: Our implementation.

| Method | #epochs | backbone | AP | $AP_r$ | $AP_c$ | $AP_f$ |
|---|---|---|---|---|---|---|
| ViLD [25] | 460 | R50 | 27.8 | 16.7 | 26.5 | 34.2 |
| DetPro [20] | 20 | R50 | 28.4 | 20.8 | 27.8 | 32.4 |
| VLDet [41] | 96 | R50 | 33.4 | 22.9 | 32.8 | 38.7 |
| BARON [63] | 24 | R50 | 29.5 | 23.2 | 29.3 | 32.5 |
| DK-DETR [38] | 70 | R50 | 33.5 | 22.2 | 32.0 | 40.2 |
| DINO-det-4scale* | 24 | R50 | 32.5 | 15.2 | 32.8 | 39.8 |
| **Frozen-DETR** (Ours) | 24 | R50 | **40.0** | **24.0** | **41.8** | **44.9** |

Table 11: Results of combining multiple foundation models on COCO.

| CLIP [54] | DINOv2 [52] | AP | $AP_{50}$ | $AP_{75}$ | $AP_s$ | $AP_m$ | $AP_l$ |
|---|---|---|---|---|---|---|---|
| | | 49.0 | 66.6 | 53.5 | 32.0 | 52.3 | 63.0 |
| ✓ | | 51.9 (+2.9) | 70.4 | 56.7 | 33.8 | 54.9 | 69.3 |
| ✓ | ✓ | **53.8** (+4.8) | **72.3** | **58.7** | **35.2** | **57.5** | **72.3** |

has an impressive improvement of 6.6% AP over DINO-det, demonstrating that the benefit of image understanding from foundation models is even more significant in the more challenging scenario. Further, the improvement on rare (+8.7% AP) and common classes (+7.7% AP) is larger than frequent classes, showing that Frozen-DETR has the potential to alleviate the class imbalance problem.

**Open-Vocabulary Setting.** As CLIP demonstrates an impressive zero-shot ability, many works [25, 20, 71] aim to inherit its generalization ability to achieve open-vocabulary recognition. In this subsection, we also validate the open-vocabulary ability inherited by Frozen-DETR. In this setting, annotations for rare classes are removed and only common and frequent class annotations are used for training. The AP on rare classes is the main evaluation metric. We follow common practices by replacing the classifier with class prompts, which are encoded by CLIP text encoder with 80 hand-crafted prompts, *e.g.*, "a photo of {category} in the scene". No other distillation methods [25, 63, 38] or additional datasets [71, 41] are used. As shown in Table 10, since we use CLIP as a frozen feature enhancer, the open-vocabulary ability is largely inherited. We outperform the baseline DINO-det by 8.8% $AP_r$. We also outperform many detectors tailored for open-vocabulary detection, even though it is not a fair comparison as we use CLIP ViT-L-14-336 and others use CLIP ViT-B-32.

### 4.5 Combining Multiple Foundation Models

In this subsection, we explore whether combining multiple foundation models can further improve the performance since image understanding abilities from different aspects may be learned by different foundation models trained with different pre-tasks. Here we try to combine DINO-det-4scale [66] with CLIP [54] and DINOv2 [52]. Both foundation models use ViT-L-14-336. We only use the patch tokens from DINOv2 as another feature map ($4 + 2 = 6$ feature maps now). As shown in Table 11, we can further increase DINO-det to 53.8% AP (+4.8% AP), showing that different foundation models may be complementary in the aspect of image understanding.

### 4.6 Transfering Frozen-DETR to Other Domains

In the real world, input images always suffer from natural distribution shifts. We also find that Frozen-DETR inherits great domain generalization ability from frozen foundation models. We directly transfer the model trained on the COCO dataset to the COCO-O dataset [50] without fine-tuning, which is a dataset having the same classes as COCO but different domains, such as sketch, weather, cartoon, painting, tattoo, and handmake. As shown in the Table 12, Frozen-DETR achieves almost the same performance on both datasets, while other detectors degrade a lot on the COCO-O. The performance of Frozen-DETR on COCO-O is two times higher than the baselines and even higher than detectors with strong backbones, showing its strong robustness.

### 4.7 How does Frozen-DETR work?

To understand how Frozen-DETR works, we conduct error analysis [7] in Table 13. The location error (Loc) denotes the predictions with correct labels but low IoUs. The classification error (Cls) denotes the predictions with the correct locations but incorrect labels. The background error (BG)

Table 12: Results on the COCO-O dataset. The models are trained on the COCO datasets and directly tested on the COCO-O dataset without finetuning. ER denotes Effective Robustness.

| Method | Backbone | COCO mAP | COCO-O (mAP) | | | | | | | ER |
| --- | --- | --- | --- | --- | --- | --- | --- | --- | --- | --- |
| | | | Sketch | Weather | Cartoon | Painting | Tattoo | Handmake | Avg. | |
| DINO-det-5scale [66] | Swin-L | 58.5 | - | - | - | - | - | - | 42.1 | +15.76 |
| ViTDet [40] | ViT-H | **58.7** | - | - | - | - | - | - | 34.3 | +7.89 |
| DETR [3] | R50 | 42.0 | 9.0 | 30.0 | 12.3 | 23.9 | 11.6 | 15.7 | 17.1 | -1.82 |
| Deformable DETR [73] | R50 | 44.5 | 10.5 | 30.2 | 15.1 | 26.2 | 10.6 | 18.6 | 18.5 | -1.49 |
| DINO-det-4scale [66] | R50 | 49.0 | 13.8 | 36.3 | 18.5 | 30.7 | 13.3 | 22.4 | 22.5 | +0.45 |
| **Frozen-DETR** (DINO-det+CLIP) | R50 | 51.9 | 50.3 | 46.3 | 51.4 | 54.9 | 52.1 | 46.0 | 50.2 | +26.8 |
| **Frozen-DETR** (DINO-det+CLIP+DINOv2) | R50 | 53.8 | **52.8** | **49.3** | **53.5** | **56.6** | **57.7** | **52.3** | **53.7** | **+29.49** |

Table 13: Error analysis of models with and without foundation models on COCO.

| Method | AP | $AP_{50}$ (acc)↑ | Loc↓ | Cls↓ | BG↓ | FN↓ |
| --- | --- | --- | --- | --- | --- | --- |
| DINO-det-4scale | 49.0 | 66.7 | 7.0 | 9.6 | 12.4 | 4.3 |
| +CLIP | 51.9 | 70.4 | 7.4 (+0.4) | 8.2 (-1.4) | 10.5 (-1.9) | 3.5 (-0.8) |
| +CLIP+DINOv2 | 53.8 | 72.3 | 7.1 (+0.1) | 7.7 (-1.9) | 9.9 (-2.5) | 3.0 (-1.3) |

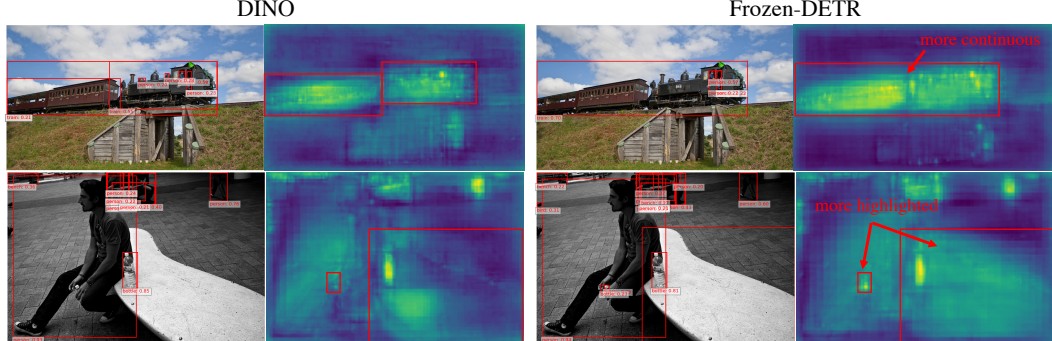

Figure 4: Predictions and feature maps from DINO [66] and Frozen-DETR (CLIP only).

indicates the detector erroneously marks a background region as an object. The false negative error (FN) means the detector overlooks some annotated objects.

The results in Table 13 validate that the benefit of CLIP and DINOv2 comes more from high-level semantic understanding rather than texture details, which **is good for classification more than localization**. With strong image understanding, the detector can find missing objects (lower false negative error) and correct wrong classification (lower classification and background error). We also visualize the feature maps ($l_2$ norm) after the encoder in Figure 4 to better illustrate the benefit. The enhanced high-level semantic understanding leads to a more complete activation of objects (the first row), ensuring that objects are detected as complete entities. Further, it allows certain objects to stand out distinctly against the background (the second row). More visualizations are provided in the Appendix.

## 5  Conclusions

In this work, we show that frozen foundation models can be versatile feature enhancers, even though they are not pre-trained for object detection. We explore a new way to utilize pre-trained foundation models as a feature enhancer rather than a backbone. Class tokens from them provide a compact context for decoding object queries in the decoder. Patch tokens further inject semantic details into feature maps via feature fusion in the encoder. Experiments show that CLIP is one of the best candidates for Frozen-DETR and the image understanding ability in CLIP can greatly enhance the classification ability of DETRs, especially in large vocabulary settings. We hope our work can shed new light on reusing the existing strong foundation models on various downstream tasks.

## Acknowledgments and Disclosure of Funding

This work was supported partially by NSFC(92470202, U21A20471), Guangdong NSF Project (No. 2023B1515040025) and the Major Key Project of PCL under Grant PCL2024A06. This work was also supported by Alibaba Innovative Research Program.

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

# A Comparisons between Using Foundation Models as a Backbone and as a Plug-and-Play Module

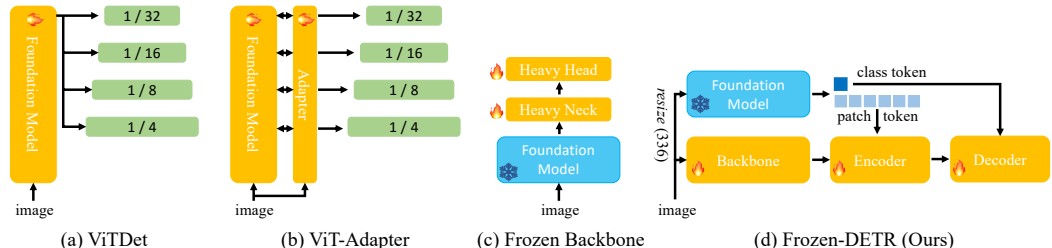

(a) ViTDet      (b) ViT-Adapter      (c) Frozen Backbone      (d) Frozen-DETR (Ours)

Figure 5: Different types of usage of pre-trained vision foundation models. (a) ViTDet [40] fully fine-tunes the whole foundation model. (b) ViT-Adapter [13] injects task priors to foundation models by adapters. Both the foundation model and adapters are fine-tuned on the downstream tasks. (c) Some works [58, 44] explore using frozen foundation models as the backbone, which needs a heavy neck and heavy head to ensure that there are enough tunable parameters. (d) Our Frozen-DETR utilizes foundation models as a plug-and-play module, in which the foundation model is not trainable and the image size is much smaller than the one in the detector.

Table 14: Comparisons with different methods to improve the performance.

| Method | Training | | Inference | | GFLOPs |
|---|---|---|---|---|---|
| | Mem | time / epoch | Mem | FPS | |
| DINO-det-4scale baseline | 13G (bs=2) | 1.3h | 3G | 9.7 | 279 |
| Frozen-DETR (DINO-det-4scale) | 15G (bs=2) | 1.4h | 3G | 6.5 | 400 |
| DINO-det-5scale | 34G (bs=2) | 2.6h | 5G | 4.4 | 860 |
| DINO-det-4scale + ViT-L backbone | 44G (bs=1) | 4.2h | 10G | 2.1 | 1244 |

In this work, we propose a novel paradigm (comparisons are shown in Figure 5) to integrate frozen vision foundation models with query-based detectors, firstly showing that frozen foundation models can be a versatile feature enhancer to boost the performance of detectors, even though they are not pre-trained for object detection.

In previous practices, large vision foundation models are always used as a pre-trained backbone and fine-tuned with detectors in an end-to-end manner. Although such a paradigm achieves high performance, the computation cost of fine-tuning such a large vision foundation model is unaffordable. We use ViT-L as an example to illustrate this problem, as ViT-L is a common architecture for most vision foundation models. In the above table, we choose DINO-det-4scale with R50 backbone as the baseline and compare it with three methods: our Frozen-DETR (CLIP ViT-L-336), DINO-det-5scale, and DINO-det-4scale with a foundation model (ViT-L) as the backbone. We use the ViT-L as the backbone following ViTDet. For the training, we use 4 A100 GPUs with 2 images per GPU except for the ViT-L backbone due to out-of-memory (OOM). For inference, we use a V100 GPU with batch size 1 in line with the main text. As shown in the Table 14, the computation cost in both training and inference for Frozen-DETR is the lowest among the three variants.

- Compared with DINO-det-4scale with a foundation model as a backbone, training a ViT-L backbone needs 4.2 hours per epoch and 44 GB memory per GPU, which is significantly higher than our Frozen-DETR (1.4 hours and 15 GB with 2 images per GPU). For inference, using ViT-L as a backbone needs 10 GB GPU memory and runs at 2.1 FPS on a V100 GPU. While inference with Frozen-DETR only needs 3 GB GPU memory (3x fewer) and runs at 6.5 FPS (3x faster).

- Compared with DINO-det-5scale, our Frozen-DETR not only runs faster but also significantly outperforms DINO-det-5scale by 1.8% AP (53.1% AP vs 51.3% AP), as shown in Table 8.

Thus, Frozen-DETR achieves a good performance-speed trade-off.

# B More Visualizations

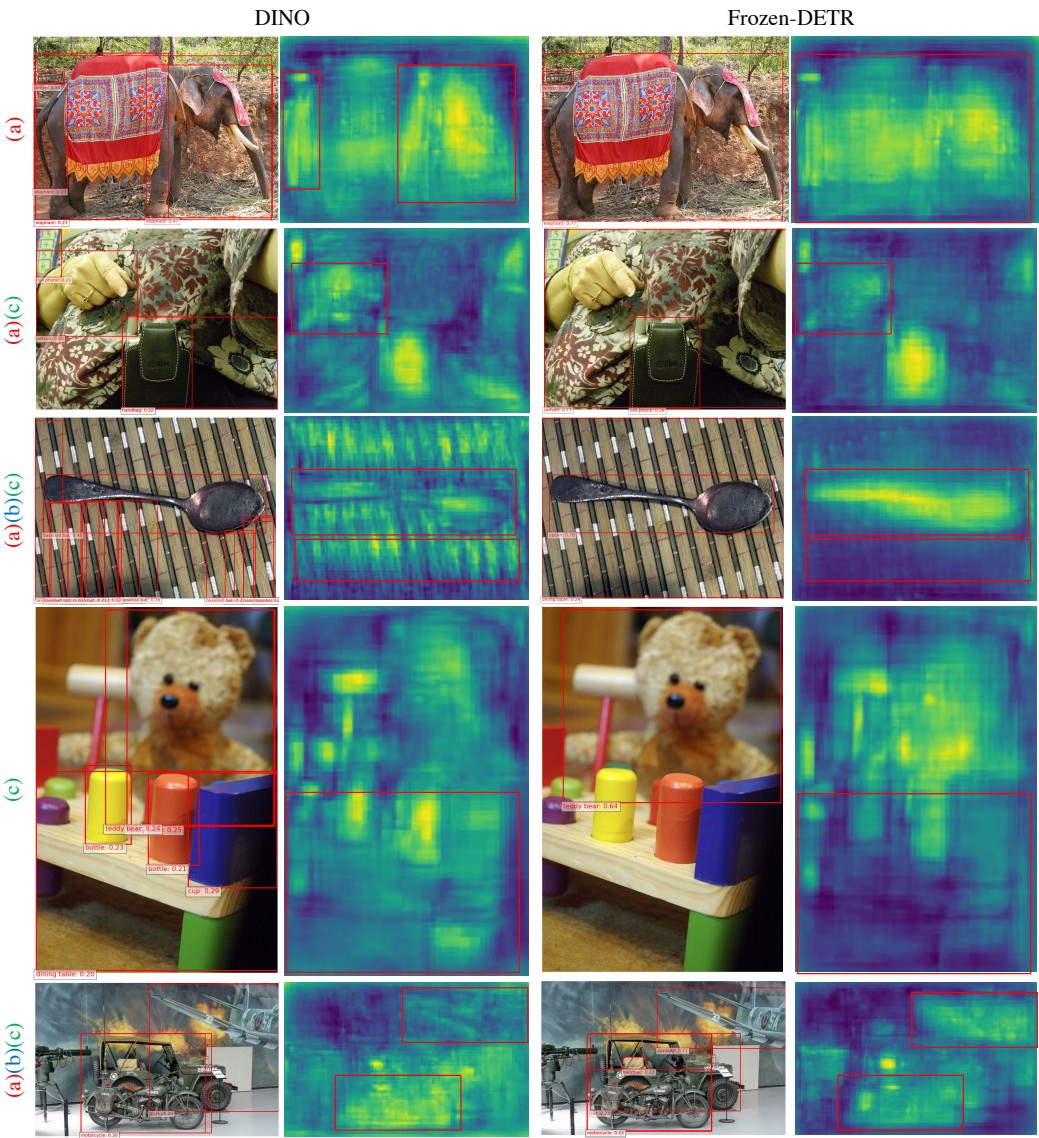

Figure 6: More visualization of the predictions and the feature maps from DINO-det-4scale [66] and Frozen-DETR (CLIP only). Using foundation models can (a) clarify the relation between parts and the whole object, (b) find missing objects, and (c) correct wrong classifications.

In Figure 6, we show more results with or without the foundation model. With the high-level image understanding ability from foundation models, the detector can detect objects as completely as possible, such as the elephant in the first image and the person in the second image. Additionally, the detector can find some missing objects, *e.g.* the strange and incomplete dining table in the third image. Further, with the foundation model, the detector can correctly classify the cell phone rather than a handbag in the second image and the toy (the toy is not the class in COCO) rather than bottles in the fourth image.

# C Do Foundation Models with Registers Further Improve Frozen-DETR?

Registers [15] are used in modern ViTs for mitigating artifacts, which are also helpful for our Frozen-DETR. Since this work only releases the checkpoint for DINOv2, the following experiments are

Table 15: Results of combining foundation models with registers.

| Method | AP | $AP_{50}$ | $AP_{75}$ | $AP_s$ | $AP_m$ | $AP_l$ |
|---|---|---|---|---|---|---|
| DINO-det-4scale [66] | 49.0 | 66.6 | 53.5 | 32.0 | 52.3 | 63.0 |
| +CLIP [54] | 51.9 | 70.4 | 56.7 | 33.8 | 54.9 | 69.3 |
| +DINOv2 [52] | 53.3 | 71.8 | 58.1 | 35.2 | 56.2 | 71.9 |
| +DINOv2-reg [15] | 53.9 | 72.4 | 58.8 | 34.8 | 57.2 | 72.2 |

conducted on DINOv2 and DINOv2 with registers (DINOv2-reg). In the Table 15, we find that using DINOv2 as the foundation model can even get better results than using CLIP, which is different from Table 2. We hypothesize there are two reasons: First, DINOv2 has both global-wise and token-wise pre-training pre-tasks. Thus the patch tokens are more informative. Further, DINOv2 ViT-L is distilled from ViT-giant, which equals a larger foundation model. Thus equiping the detector with DINOv2 gets higher performance. Further, we find that DINOv2-reg can mitigate artifacts in DINOv2 and further improve the performance.

# D  Limitations and Broader Impacts

**Limitations.** This work utilizes the image understanding ability in frozen foundation models. However, these models are trained on nature images and may not perform well in many challenging scenarios, *e.g.*, medical images. Although Frozen-DETR enjoys an asymmetric input size, which greatly reduces computation costs, it still slows down the detector. Distilling Frozen-DETR to a standard detector may solve the problem and preserve its high performance.

**Broader Impacts.** This work improves the existing SOTA DETR-like detectors, which can be applied to automatic driving systems and many other downstream tasks.

