# OpenReview forum: "Frozen-DETR: Enhancing DETR with Image Understanding from Frozen Foundation Models"
_NeurIPS.cc/2024/Conference — NeurIPS 2024 poster_

### Official Review · Reviewer_C9T7 · 2024-07-07

**Soundness:** 3
**Presentation:** 2
**Contribution:** 3
**Rating:** 6
**Confidence:** 5

**Summary:**

This paper proposes Frozen DETR, which leverages frozen foundation models as feature enhancers to improve the performance of the DETR object detection framework. By integrating feature maps from models like CLIP into the pyramid feature maps and feeding them into the encoder, Frozen DETR enriches the contextual information of objects within the original pyramid features, thereby enhancing DETR's performance.

**Strengths:**

1. The paper use fundation models as plug-and-play modules, which is easy to apply on different detectors and enhance the performance significantly.
2. The paper conducts extensive experiments and comparisons with the SOTA methods on different datasets to show the effectiveness and robustness.

**Weaknesses:**

1. The use of foundation models increases inference costs. It would be beneficial to provide a detailed analysis of the computational cost introduced by these models.
2. The performance improvement appears to diminish when using more advanced models like Co-DETR, especially during extended training periods such as 24 or 36 epochs. An explanation for this phenomenon would be helpful.
3. The method seems applicable to other SOTA models, such as [1,2,3]. A comparison with these models would strengthen the evaluation of the proposed approach.

[1] Zhao, J., Wei, F., & Xu, C. (2024). Hybrid Proposal Refiner: Revisiting DETR Series from the Faster R-CNN Perspective. In Proceedings of the IEEE/CVF Conference on Computer Vision and Pattern Recognition (pp. 17416-17426).

[2] Zhao, C., Sun, Y., Wang, W., Chen, Q., Ding, E., Yang, Y., & Wang, J. (2024). MS-DETR: Efficient DETR Training with Mixed Supervision. In Proceedings of the IEEE/CVF Conference on Computer Vision and Pattern Recognition (pp. 17027-17036).

[3] Wang, Y., Li, X., Weng, S., Zhang, G., Yue, H., Feng, H., ... & Ding, E. (2024). KD-DETR: Knowledge Distillation for Detection Transformer with Consistent Distillation Points Sampling. In Proceedings of the IEEE/CVF Conference on Computer Vision and Pattern Recognition (pp. 16016-16025).

**Questions:**

1. Why is the performance (47.0 AP) of the Co DINO baseline in Tables 12 and 13 inconsistent with the main paper, which reports 52.1 AP?
2. Previous work [1] noted that features from CLIP's vision encoder can have artifact problems, potentially resulting in noisy feature maps. Does this issue influence the performance of detector? If so, would applying the methods from [1] help mitigate these artifacts and improve performance?

[1] Darcet, T., Oquab, M., Mairal, J., & Bojanowski, P. (2023). Vision transformers need registers. arXiv preprint arXiv:2309.16588.

**Limitations:**

1. As mentioned in the paper, the performance gains heavily depend on the quality and characteristics of the pre-trained foundation models. Variations in these models could lead to inconsistencies and affect the reliability of the results.
2. Using foundation models increases inference inefficiency. Solutions such as knowledge distillation might help mitigate this issue and improve efficiency.

---

> ### Author Rebuttal · Authors · 2024-08-07
>
> # To Reviewer C9T7
>
> **Q1: It would be beneficial to provide a detailed analysis of the computational cost.**
>
> **RE**: We provide analyses of the computational cost in Table 3, Table 4, and Table 5 in the main text. Besides, more discussion can be found in the rebuttals for all reviewers above.
>
> ***
>
> **Q2: The performance improvement appears to diminish when using more advanced models like Co-DETR, especially during extended training periods such as 24 or 36 epochs.**
>
> **RE**: A significant advantage of most SOTA query-based detectors is that they can converge extremely fast within 12 epochs. As a result, the performance gain may not be that obvious within a long training schedule. This phenomenon is also reported by Hybrid Proposal Refine [a], where DDQ with HPR merely gains +0.1% AP improvement (from 52.4% to 52.5%) as the training epochs increase from 12 to 24. Some training strategies are excepted to further improve the performance with longer training schedules, e.g., large-scale jitter used in [a] and other data augmentation. We will add relative discussions in the revision.
>
> [a] Hybrid Proposal Refiner: Revisiting DETR Series from the Faster R-CNN Perspective, In CVPR 2024.
>
> ***
>
> **Q3: Apply Frozen-DETR to other SOTA models, such as MS-DETR, Hybrid Proposal Refiner and KD-DETR.**
>
> **RE**: Thanks for your advice. However, we would like to highlight that Hybrid Proposal Refiner and KD-DETR are not publicly available when submitting our work. These two papers are only available on CVPR2024 openaccess after 13th June. And KD-DETR is not a new detector but a distillation method. As a result, we provide the experiments on MS-DETR and Hybrid Proposal Refiner below. As shown in the table, our method can be easily applied to recent SOTA models.
>
> | Model                      | AP   | AP$_{50}$ | AP$_{75}$ | AP$_{s}$ | AP$_m$ | AP$_l$ | GFLOPs | FPS  |
> |----------------------------|------|-----------|-----------|----------|--------|--------|--------|------|
> | MS-DETR                    | 50.0 | 67.3      | 54.4      | 31.6     | 53.2   | 64.0   | 252    | 10.8 |
> | Frozen-DETR (MS-DETR)      | 53.0 | 71.5      | 57.8      | 35.1     | 55.8   | 70.8   | 452    | 6.9  |
> | DDQ with HPR               | 52.4 | 69.9      | 57.5      | 35.9     | 55.5   | 66.7   | 283    | 6.5  |
> | Frozen-DETR (DDQ with HPR) | 55.7 | 73.9      | 61.3      | 38.4     | 58.8   | 72.3   | 467    | 5.2  |
>
> ***
>
> **Q4: Why is the performance of the Co DINO baseline in Tables 12 and 13 inconsistent with the main paper?**
>
> **RE**: The baseline in Tables 12 and 13 is the same baseline as in Section 4.2, which uses four-scale feature maps and does not use co-heads. Due to space limitations of the main text, these two tables are presented in the appendix. We will add more experimental details.
>
> ***
>
> **Q5: Previous work noted that features from CLIP's vision encoder can have artifact problems. Would applying the methods from it help mitigate these artifacts and improve performance?**
>
> **RE**: Thanks for your advice. Since this work only releases the checkpoint for DINOv2-FM. Thus the following experiments are conducted on DINOv2-FM and DINOv2-FM with registers (DINOv2-reg). In the table, we find that using DINOv2 can even get better results than using CLIP, which is different from Table 2. We hypothesize there are two reasons: First, DINOv2-FM has both global-wise and token-wise pre-training pre-tasks. Thus the patch tokens are more informative. Further, DINOv2-FM ViT-L is distilled from ViT-giant, which equals to a larger foundation model. We find that DINOv2-reg can mitigate artifacts in DINOv2-FM and further improve the performance. We will add these results to the manuscript.
>
> | Model                        | AP   | AP$_{50}$ | AP$_{75}$ | AP$_{s}$ | AP$_m$ | AP$_l$ |
> |------------------------------|------|-----------|-----------|----------|--------|--------|
> | DINO-det-4scale              | 49.0 | 66.6      | 53.5      | 32.0     | 52.3   | 63.0   |
> | DINO-det-4scale + CLIP       | 51.9 | 70.4      | 56.7      | 33.8     | 54.9   | 69.3   |
> | DINO-det-4scale + DINOv2-FM  | 53.3 | 71.8      | 58.1      | 35.2     | 56.2   | 71.9   |
> | DINO-det-4scale + DINOv2-reg | 53.9 | 72.4      | 58.8      | 34.8     | 57.2   | 72.2   |
>
> ***
>
> **Q6: The performance gains heavily depend on the quality and characteristics of the pre-trained foundation models.**
>
> **RE**: Since different vision foundation models are pre-trained with different pre-tasks and Frozen-DETR does not train the vision foundation models, it is quite normal and foreseeable that the choice of vision foundation models will affect the performance, and not all of them are suitable for Frozen-DETR. We have tested and compared many representative foundation models with various pre-training methods in Table 2 with some intuitive explanations. These empirical practices can be used as the selection guidelines.
>
> ***
>
> **Q7: Solutions such as knowledge distillation might help improve efficiency.**
>
> **RE**: Thanks for your instructive advice. In recent distillation methods, the teacher model and student model should always be trained with the same task and use similar architectures. Since foundation models are not pre-trained for detection, how to transfer the knowledge from them to detectors is still an open question. We also notice that some open-vocabulary methods distill the knowledge from CLIP. However, they can not improve the base class performance from distillation or even pursue high novel class performance at the price of low base class performance [1,2,3]. While our method can easily apply to many query-based detectors, boosting all classes' performance.
>
> [1] Open-vocabulary object detection via vision and language knowledge distillation. In ICLR, 2022.
>
> [2] Distilling detr with visual-linguistic knowledge for open-vocabulary object detection. In ICCV, 2023.
>
> [3] Aligning bag of regions for open-vocabulary object detection. In CVPR, 2023.

---

> > ### Comment · Reviewer_C9T7 · 2024-08-11
> >
> > Thanks for the rebuttal. The new experiments and answers have mostly addressed my concerns and really show the effectiveness of frozen-DETR. I hope to see a more complete version soon. Based on this, I've decided to increase the rate.

---

### Official Review · Reviewer_QbuY · 2024-07-12

**Soundness:** 3
**Presentation:** 3
**Contribution:** 2
**Rating:** 6
**Confidence:** 5

**Summary:**

This paper incorporates frozen foundation model backbones into DETR pipelines. Specifically, the paper concatenates the output of the class token of the foundation models with the query vector in the decoder. Also, it concatenates the patch tokens with the feature pyramid from the Image backbone to improve accuracy.

**Strengths:**

1. The paper is well written.
2. The major novelty is fusing features of the frozen foundation models with backbone output and the query vectors in the decoder.
3. The idea of using reduced resolution for the foundation model is engaging in a sense to reduce the overall compute.

**Weaknesses:**

Despite the paper's merits, it also has weaknesses:

1. The idea of feature fusion seems incremental from the perspective of technical contributions. By looking at the design, the feature fusion of the frozen foundation model is the only principle claim of the paper. Apart from that, I could not see any other piece of substantial advancement.

2. Frozen-DETR is only tested with DINO, which has huge computational costs due to dense multiscale feature utilization in the encoder. To demonstrate the utility of the presented idea, there should be results with affordable DETRs, e.g. DN-DETR, Conditional-DETR, DAB-DETR, Anchor-DETR, IMFA -DETR, Deformable-DETR etc. These are the pivotal works in this area, and hence, evaluation of the proposed Frozen-DETR pipeline is a must for this paper, given its limited technical contributions.

3. I am more concerned about the applicability of this method in the real world, considering its weaker accuracy improvement tradeoffs vs runtime. By looking at Tables 12 and 13, changing the image resolution does not aggressively improve the accuracy. For example, in Table 12, doubling the resolution of the foundation model halves the speed where the original speed is already very low while the accuracy improvement is 1.8AP. Similar effects are seen in Table 13. Due to this reason, the weakness in point 2 (see above) must be addressed.

**Questions:**

1. Include the results of pivotal DETR methods instead of only one DINO, which is computationally heavy.
2. Provide detailed runtime of each model to assess better the work given limited technical contributions.

If this is resolved, I am open to adjusting my score.

**Limitations:**

Please refer to the weakness section.

---

> ### Author Rebuttal · Authors · 2024-08-07
>
> # To Reviewer QbuY
>
> **Q1: The idea of feature fusion seems incremental from the perspective of technical contributions. By looking at the design, the feature fusion of the frozen foundation model is the only principle claim of the paper.**
>
> **RE**: We respectfully disagree with this point. We would like to highlight the core contribution of this work, which is a novel paradigm to integrate frozen vision foundation models with query-based detectors. We are the first to show that frozen foundation models can be a versatile feature enhancer to boost the performance of detectors **without any fine-tuning, even though they are not pre-trained for object detection**. Such a paradigm enjoys many advantages over using vision foundation models as the backbone. Please see the rebuttals for all reviewers above for more details.
>
> ***
>
> **Q2: Frozen-DETR is only tested with DINO, which has huge computational costs due to dense multiscale feature utilization in the encoder. To demonstrate the utility of the presented idea, there should be results with affordable DETRs, e.g. DN-DETR, Conditional-DETR, DAB-DETR, Anchor-DETR, IMFA -DETR, Deformable-DETR etc.**
>
> **RE**: We have applied the method to AdaMixer, DINO-det, and Co-DETR in the paper, which can demonstrate the generalization ability. We have also applied our Frozen-DETR to MS-DETR and HPR (See Reviewer C9T7 Q3 below). Due to limited time and computation resources, we select DN-DETR and DAB-DETR as the representatives of single-scale query-based detectors. Both models are trained with 12 epochs. Experimental results show that Frozen-DETR can also significantly enhance the performance of single-scale detectors (+ 4.0% AP on DAB-DETR and + 2.7% AP on DN-DETR) with acceptable additional computation cost.
>
> | Model                      | AP   | AP$_{50}$ | AP$_{75}$ | AP$_{s}$ | AP$_m$ | AP$_l$ | GFLOPs | FPS  |
> |----------------------------|------|-----------|-----------|----------|--------|--------|--------|------|
> | DAB-DETR-DC5               | 38.0 | 60.3      | 39.8      | 19.2     | 40.9   | 55.4   | 220    | 10.2 |
> | Frozen-DETR (DAB-DETR-DC5) | 42.0 | 63.2      | 44.9      | 22.4     | 45.4   | 61.1   | 372    | 8.5  |
> | DN-DETR-DC5                | 41.7 | 61.4      | 44.1      | 21.2     | 45.0   | 60.2   | 220    | 10.2 |
> | Frozen-DETR (DN-DETR-DC5)  | 44.4 | 64.8      | 47.7      | 23.8     | 47.7   | 64.6   | 372    | 8.5  |
>
> We also notice that the models can be further accelerated without using DC5 but with lower performance (lags behind around 2% AP). Without DC5, DN-DETR has 104 GFLOPs and runs at 21.0 FPS, while Frozen-DETR has 298 GFLOPs and runs at 14.1 FPS. Unfortunately, these single-scale models lag behind SOTA models by more than 10\% AP and need 50 epochs to converge. Thus, these models can not meet practical needs.
>
> ***
>
> **Q3: I am more concerned about the applicability of this method in the real world, considering its weaker accuracy improvement tradeoffs vs runtime. By looking at Tables 12 and 13, changing the image resolution does not aggressively improve the accuracy. Similar effects are seen in Table 13.**
>
> **RE**: We believe the accuracy improvement of Frozen-DETR is significant and all other reviewers agree with us on the promising performance:
> - On the COCO dataset, we increase 2.9% AP for DINO-det (Table 6).
> - On the challenging large vocabulary LVIS dataset, we increase 6.6% AP for DINO-det (Table 7).
> - On the challenging long-tail scenario, we increase 8.7% APr and 7.7% APc for DINO-det (Table 7), showing the potential to alleviate the class imbalance problem.
> - On the challenging open-vocabulary scenario, we increase 8.8% novel AP for DINO-det (Table 8), showing strong open-vocabulary ability.
> - In the real world, input images always suffer from natural distribution shifts. We also find that Frozen-DETR inherits great domain generalization ability from frozen foundation models. We directly transfer the model trained on the COCO dataset to the COCO-ood dataset [1] without fine-tuning, which is a dataset having the same classes as COCO but different domains, such as sketch, weather, cartoon, painting, tattoo, and handmake. As shown in the table below, Frozen-DETR achieves almost the same performance on both datasets, while other detectors degrade a lot on the COCO-O. The performance of Frozen-DETR on COCO-O is two times higher than the baselines and even higher than detectors with strong backbones, showing its strong robustness. We will add these results to the appendix.
>
> | Model                                     | Backbone | COCO AP | COCO-O AP |
> |-------------------------------------------|----------|---------|-----------|
> | DINO-det                                  | Swin-L   | 58.5    | 42.1      |
> | ViTDet                                    | ViT-H    | 58.7    | 34.3      |
> |-------------------------------------------|----------|---------|-----------|
> | DETR                                      | R50      | 42.0    | 17.1      |
> | Deformable DETR                           | R50      | 44.5    | 18.5      |
> | DINO-det                                  | R50      | 49.0    | 22.5      |
> |-------------------------------------------|----------|---------|-----------|
> | Frozen-DETR (DINO-det + CLIP)             | R50      | 51.9    | 50.2      |
> | Frozen-DETR (DINO-det + CLIP + DINOv2-FM) | R50      | 53.8    | 53.7      |
>
> Further, in Table 12, we aim to validate the property that Frozen-DETR enjoys the asymmetric input size. Using a small image resolution is already feasible to achieve promising results thus achieving a good performance-speed trade-off. In Table 13, we aim to demonstrate that a stronger foundation model can obtain larger improvements. Please see the rebuttals for all reviewers above for more discussions on the performance-speed trade-off.
>
> [1] COCO-O: A Benchmark for Object Detectors under Natural Distribution Shifts. In ICCV 2023.

---

> > ### Comment · Reviewer_QbuY · 2024-08-11
> > **Response to Authors**
> >
> > I thank the authors for providing additional experimentation.
> >
> > I am still not convinced of the novelty. I agree that this paper is the first to fuse the features of a foundation model with DETR queries however this is not a very big technological contribution itself.
> >
> > Incorporating more information from foundation models will be beneficial and the same has been demonstrated in the paper. I agree with the improved detection performance but not with the runtime performance as they are not convincing. The primary reason is itself has been shown in the author's rebuttal that even single scale DN-DETR has 108G Flops whereas Frozen-DETR has 298G FLOPS which is three times.
> >
> >  I have a strong opinion that your FPS calculations are done on a pretty advanced GPU with a huge number of cores hence the FPS gap is visible but not as visible as compared to the FLOPs difference. So the difference of roughly 200GF will become more prominent when GPU is a commodity version.
> >
> > Moreover, I don't see FLOP values except in Table 4 which says your model crosses 400G FLOPs which is humongous.
> >
> > In the main tables such as Table 6,  FLOPs are not mentioned. Also as per line 219, the FLOPs contributed significantly due to the foundation model.
> >
> > Hence, I encourage authors to provide the exact FLOPs of each method in Table-6 while the FPS of as many methods because it would be crucial for my final rating.

---

> ### Author Response · Authors · 2024-08-12
> **Response to Reviewer QbuY**
>
> | Detector   | Multi-scale? | Encoder? | \# Epochs | AP | GFLOPs | FPS (V100) | FPS (1080Ti) |
> |-------------|--------------|----------|-----------|------|--------|------------|--------------|
> | DETR | $\times$ |  | 500 | 43.3 | 86 | 27.8 | 21.6 |
> | Deformable DETR | | | 50 | 43.8 | 173 | 13.4 | 8.8 |
> | Sparse R-CNN | | x | 36  | 45.0 | 174 | 17.8 | 13.6 |
> | AdaMixer |  | x | 36 | 47.0 | 132 | 16.6 | 11.8 |
> | DDQ DETR 4scale |  |  | 24 | 52.0 | 249 | 8.6 | 5.9 |
> | Group DETR (DINO 4scale) |  |  | 36 | 51.3 | 279 | 9.7 | 6.7 |
> | H-Deformable-DETR | | | 36 | 50.0 | 268 | 11.0 | 5.6 |
> | DAC-DETR | | | 24 | 51.2 | 279 | 9.7 | 6.7 |
> | DAB-DETR-DC5 | x | | 12 | 38.0 | 220 | 10.2 | 5.0 |
> |**Frozen-DETR (DAB-DETR-DC5)** | x | | 12 | 42.0 | 372 | 8.5 | 4.7 |
> | DN-DETR-DC5 | $\times$ | | 12 | 41.7 | 220 | 10.2 | 5.0 |
> | **Frozen-DETR (DN-DETR-DC5)** | x | | 12 | 44.4 | 372 | 8.5 | 4.7 |
> | DINO 4scale | | | 12 | 49.0 | 279 | 9.7 | 6.7 |
> | DINO 4scale | | | 24 | 50.4 | 279 | 9.7 | 6.7 |
> | DINO 5scale | | | 24 | 51.3 | 860 | 4.4 | 2.4 |
> | **Frozen-DETR (DINO 4scale)**| | | 12 | 51.9 | 400 | 6.5 | 4.3 |
> | **Frozen-DETR (DINO 4scale)** | | | 24 | 53.2 | 400 | 6.5 | 4.3 |
> | MS-DETR | | | 12 | 50.0 | 252 | 10.8 | 6.5 |
> | **Frozen-DETR (MS-DETR)** | | | 12 | 53.0 | 452 | 6.9 | 4.3 |
> | DDQ with HPR | | | 12 | 52.4 | 283 | 6.5 | 4.4 |
> | **Frozen-DETR (DDQ with HPR)**   | | | 12 | 55.7 | 467 | 5.2 | 3.3 |
> | Co-DINO 5scale | | | 12 | 52.1 | 860 | 4.4 | 2.4 |
> | **Frozen-DETR (Co-DINO 4scale)** | | | 12 | 52.8 | 400 | 6.5 | 4.3 |
> | **Frozen-DETR (Co-DINO 4scale)** | | | 24 | 53.5 | 400 | 6.5 | 4.3 |
>
> Thanks for your feedback and agreement on two key contributions of our work: 1) the first attempt to apply frozen foundation models in the downstream object detection task and 2) the improved detection performance.
>
> For the technological contribution, we design a decoupled feature fusion method by **viewing each component of the foundation models as a special part of detectors**, such as class tokens as image queries and patch tokens as another level of feature pyramid. Such a feature fusion technique can effectively transfer the image understanding ability from foundation models to detectors with minimal modifications to detectors so that the method can be easily applied to various query-based detectors. Considering the various benefits of our Frozen-DETR compared to other ways to use foundation models in detection discussed before, we hope that the Frozen-DETR paradigm can arouse researchers' interest in introducing foundation models into detection.
>
> For the computation cost concern, we provide the detailed FLOPs and FPS for all the models in the above table. We would like to highlight some key points:
> - Our Frozen-DETR can be applied to various query-based detectors, including both single-scale and multi-scale detectors. The inference time is within 1.5x the time of baselines, which is acceptable considering the performance improvement. **Reviewer RksL agrees with us**.
> - The detectors without multi-scale feature maps or encoders run faster but **lag behind SOTA models by a clear margin** (for example, DAB-DETR-DC5 only gets 38.0\% AP with training 12 epochs). To get better performance, they should be equipped with a stronger backbone or other modules, which will also slow down the speed. While our Frozen-DETR can achieve high performance, especially in many challenging scenarios mentioned in the rebuttals above.
> - As discovered by many papers, FPS does not strictly increase in proportion to FLOPs, as many factors also influence the FPS, such as the complexity of operators, memory access cost (MAC), and GPU parallel computing utilization [1]. For example, Deformable DETR and Sparse RCNN have similar FLOPs but with different fps (13.4 vs 17.8), as shown in the above table. Since foundation models enjoy a simple architecture and Attention is highly optimized for modern GPUs, most operators can be computed in parallel. The fps gap is smaller than that of FLOPs. We recommend to use FPS as the main metric for computation cost.
> - Further, many inference-time acceleration methods [2-5] for foundation models have been proposed, which can further accelerate Frozen-DETR. Besides, Frozen-DETR can be equipped with different foundation models. A faster foundation model can also accelerate Frozen-DETR.
> - According to your comments, we also change the V100 GPU to a slower 1080 Ti GPU. The trend in FPS is the same for different types of GPU.
>
> [1] An Energy and GPU-Computation Efficient Backbone Network for Real-Time Object Detection. In CVPRW 2019.
>
> [2] FasterTransformer. By NVIDIA.
>
> [3] TensorRT-LLM. By NVIDIA.
>
> [4] Flashattention: Fast and memory-efficient exact attention with io-awareness. In NeurIPS 2022.
>
> [5] Efficient Memory Management for Large Language Model Serving with PagedAttention. In SOSP 2023.

---

> > ### Comment · Reviewer_QbuY · 2024-08-12
> > **Response to the Authors**
> >
> > Thank you for the responses and for providing more FLOP details.
> >
> > I'm afraid I have to disagree with the FPS trend as seen in the 1080Ti. This 1080Ti is weaker than the V100 GPU and has far fewer cores.
> >
> > For example, DAB-DETR-DC5/Frozen-DETR (DAB-DETR-DC5) achieved 10.2/8.5 FPS on V100 and 5/4.7 FPS on 1080Ti.
> > Considering the large compute requirements introduced by the Frozen foundation models and roughly 150G Flops additional requirement, the FPS gap on 1080Ti should be far more than the gap on V100.
> >
> > Hence I strongly encourage the authors to double-check the values.

---

> ### Author Response · Authors · 2024-08-12
> **Response to Reviewer QbuY**
>
> Thanks for your quick reply!
>
> We have checked the code (the code is modified from benchmark.py from Deformable-DETR and we will release our code in the future) and tested the fps for DAB-DETR-DC5/Frozen-DETR (DAB-DETR-DC5) on 1080ti three more times and got the same results:
>
> DAB-DETR-DC5: 4.9, 4.9, 5.0
>
> Frozen-DETR (DAB-DETR-DC5): 4.7, 4.7, 4.7
>
> We also provide the fps for DAB-DETR-DC5/Frozen-DETR (DAB-DETR-DC5) on various types of GPUs: 1080Ti (5.0/4.7), P100(6.0/5.4), 2080Ti (8.2/7.6), V100 (10.2/8.5), 3090 (12.6/11.8), A100 (21.4/19.9). The fps gaps are similar across various GPUs. We find that the GPU utilization in 1080Ti is around 90% and around 70% in V100 and Frozen-DETR has a higher GPU utilization rate than baselines. We hypothesize that the GPU utilization rate is one of the reasons for the similar fps gaps across different GPUs and small fps gaps compared with FLOPs gaps. As mentioned above, many other factors will also influence the FPS.
>
> Once again, we highly recommend you select fps (the real runtime) as the main metric instead of flops. The runtime can be further improved by many off-the-shelf acceleration techniques.

---

> ### Comment · Reviewer_QbuY · 2024-08-13
> **Response to the authors**
>
> Thank you for the additional results.
>
> However, I am still unconvinced with GPU FPS performance across GPUs. They seem erratic. The primary reason is that the Frozen foundation model is run sequentially (i.e. before the main pipeline backbone or right after the main pipeline backbone). Hence 1080Ti like GPUs should technically have more FPS gap which is contrary to the evaluations provided by the authors. The current evaluations are showing 4.9FPS without the frozen model and 4.7FPS with the frozen model.
>
> Understanding, in terms of latency, 4.9FPS is 204ms while 4.7FPS is 212ms i.e. a gap of 8ms. In fact, with a stronger GPU this latency should decrease and hence on the stronger GPU, the FPS gap should be smaller. Given the size of the frozen model (DINO (transformer-based or any other)), the latency introduced by the frozen model of 8ms on 1080Ti is not justifiable in any case. Moreover, with these values, 204ms was introduced by the main detection pipeline while 8ms by the frozen foundation model which is only 3%. This directly counters your response of "Frozen-DETR has a higher GPU utilization rate".
>
>
> Considering the above argument, I'll keep my original rating.

---

> > ### Author Response · Authors · 2024-08-13
> > **Response to Reviewer QbuY**
> >
> > To Reviewer QbuY:
> >
> > Thanks for providing more details for your judgment.
> > We find that there may **be a misunderstanding** about our Frozen-DETR.
> > The single-scale DAB-DETR-DC5 uses standard self-attention in the encoder by default.
> > Meanwhile, Frozen-DETR (DAB-DETR-DC5) introduces another level of the feature pyramid from the foundation model, thus forming multi-scale feature maps.
> > Following the common design of multi-scale query-based detectors, we replace the standard self-attention in the encoder with deformable attention.
> > This difference only occurs when Frozen-DETR is applied to a single-scale query-based detector since Frozen-DETR is naturally a multi-scale detector.
> > Thus, the latency introduced by the frozen foundation model on 1080Ti cannot be simply calculated as 212ms - 204ms = 8ms.
> >
> > Here we provide detailed latency for each component of the model.
> > - DAB-DETR-DC5: 5.0FPS, 200ms on 1080Ti and 10.2FPS, 98ms on V100.
> > - Frozen foundation model: 13.2FPS, 76ms on 1080Ti and 22.2FPS, 45ms on V100.
> > - Frozen-DETR (DAB-DETR-DC5) but using random tensors as the output of the foundation model (do not forward the foundation model): 8FPS, 125ms on 1080Ti and 13.8FPS, 72ms on V100.
> > - Frozen-DETR (DAB-DETR-DC5): 4.7FPS, 213ms on 1080Ti and 8.5FPS, 118ms on V100.
> >
> > As for the FPS gaps across different GPUs, we think they are related to the GPUs' architecture.
> >
> > As mentioned in your reply, we find that our Frozen-DETR can be **further accelerated by parallel computing**. Since the foundation model and the backbone compute independently. We can simply add two lines of code `s = torch.cuda.Stream()` and `with torch.cuda.stream(s)` to make different cuda streams to execute the computation in parallel. Based on it, Frozen-DETR (DAB-DETR-DC5) can be accelerated from **4.7 FPS to 5.2 FPS** on 1080Ti and from **8.5 FPS to 8.9 FPS** on V100. We believe the code can be further optimized and many off-the-shelf acceleration techniques can be used.
> >
> > Last but not least, we will release all the experimental code, including both the main text and the rebuttals, to ensure reproducibility.
> > If you have any questions, please feel free to contact us. We are pleased to answer them.

---

> > > ### Comment · Reviewer_QbuY · 2024-08-13
> > > **Response to the authors**
> > >
> > > This is exactly my point. As you have said, 212ms - 204ms = 8ms does not make sense but your reported number says so.
> > >
> > > As acknowledged by the authors via runtime of each component, the latency introduced by the foundation model is 76ms while only DAB-DETR-DC5 is 125ms in the Frozen DETR pipeline which contrasts with the claims of your recent argument that "Frozen-DETR (DAB-DETR-DC5) introduces another level of the feature pyramid from the foundation model, thus forming multi-scale feature maps. ". Hence, the cost should be even more.
> > >
> > > Now It is confusing that why "DAB-DETR-DC5: 5.0FPS, 200ms on 1080Ti " and why "Frozen-DETR (DAB-DETR-DC5): 8FPS, 125ms on 1080Ti" holds. These two values provided by you are quite contrastive because the Foundation model is not available in both cases. DAB-DETR-DC5 does not have multiscales but the Frozen-DETR variant has multiscale features. Hence, the latency should be lower in "DAB-DETR-DC5" as compared to "Frozen-DETR (DAB-DETR-DC5) (not forwarding the foundation model)" but it is exactly the opposite.
> > >
> > > Lastly, separating two computations in cuda.Stream() is not guaranteed to be parallelized because it entirely depends on the GPU cores' availability at that instant. Hence if one component e.g. your main backbone has already occupied GPU cores, in that case, cuda.stream() will have no effect.

---

> ### Author Response · Authors · 2024-08-14
> **Response to Reviewer QbuY**
>
> To Reviewer QbuY:
>
> Thanks for your patience and quick reply. We understand that the newly added DAB-DETR-DC5 experiment during the rebuttal, with its slightly different setup, may cause some confusion. We appreciate the opportunity to clarify any misunderstandings, and all clarifications will be incorporated into the main text.
>
> As mentioned in our last reply, all single-scale detectors use standard dense self-attention in the encoder, which has the quadratic computation w.r.t. token numbers. Taking DAB-DETR-DC5 as an example, for an image with the input size 800*1200, the token numbers in the encoder will be (800 / 16) * (1200 / 16) = 3750.
>
> In multi-scale detectors, it is unreasonable to concatenate all the multi-scale feature maps along the token number dimension because there are too many tokens for multi-scale features. Thus, all multi-scale query-based detectors use deformable attention in the encoder, which is sparse attention with linear complexity to approximate the standard self-attention. Our frozen-DETR (DAB-DETR-DC5) has multi-scale feature maps and uses deformable attention in the encoder, following common practices in multi-scale query-based detectors. **In summary**, the foundation model in Frozen-DETR introduces extra latency, while deformable attention reduces latency. This results in an insignificant FPS gap between DAB-DETR-DC5 and our Frozen-DETR, which is the reason for your confusion.
>
> Regarding parallel computing, we have demonstrated that it does accelerate our Frozen-DETR even on a 1080Ti GPU. Besides, many researchers are now focusing on the efficient inference for large foundation models and LLMs. We believe these techniques could also be effectively applied to our Frozen-DETR.
>
> Considering the various advantages of Frozen-DETR, especially under many challenging scenarios (which you also acknowledged regarding the improved performance), we note that **all other reviewers have reached a consensus with us on the good performance-speed trade-off of Frozen-DETR** and have given us a positive rating. We are looking forward to your feedback.

---

### Official Review · Reviewer_RksL · 2024-07-13

**Soundness:** 3
**Presentation:** 3
**Contribution:** 3
**Rating:** 6
**Confidence:** 5

**Summary:**

This paper focuses on enhancing the performance of query-based object detection models. By inserting a foundation model into the DETR framework and treating it as a plug-and-play module instead of a backbone, the performance of query-based detectors can be significantly improved. The detection performance of DETR (DINO) on COCO is substantially enhanced by inserting patch tokens into the DETR encoder, and class tokens into the DETR decoder. Moreover, since the inserted CLIP is frozen, smaller detectors can now be equipped with larger foundation models to boost efficiency. The authors have conducted extensive ablation studies to demonstrate the effectiveness and justifiability of the proposed method.

**Strengths:**

1. Through a decoupled design, the method can accept asymmetric input sizes, which greatly reduces computational load and allows smaller detectors to be paired with larger foundation models while maintaining an acceptable computation burden.

2. The experiments are very comprehensive, essentially validating every design choice and modification proposed, and thoroughly ablating multiple feasible schemes for inserting CLIP into the model.

3. The results include large vocabulary and open vocabulary tests, demonstrating the advantages of utilizing the frozen CLIP.

**Weaknesses:**

In the experiments presented in Tables 1 and 2, the detector utilizes an R50 backbone, but most of the inserted foundation models are ViT-Ls. This design introduces a larger image encoder to provide extra information, thereby effectively creating a model ensemble to enhance the knowledge of the R50 backbone. However, as the size of the detector backbone increases— for instance, when switching from R50 to Swin-Large or even ViT-Large—does this approach of inserting a frozen CLIP still lead to noticeable improvement? The paper only provides results for the largest backbone (Swin-Base) in Table 6. I believe that more results on larger backbones is needed for further discussion to verify the potential of this method.

**Questions:**

In the Supplementary Material's Table 13, the authors examine the impact of varying model sizes of the foundation model on Co-DINO. I am still curious whether the R50-CLIP would boost the performance of an R50 backbone detector. Does this increase necessitate that the model size of CLIP backbone surpasses that of the detector backbone?

**Limitations:**

Yes

---

> ### Author Rebuttal · Authors · 2024-08-07
>
> # To Reviewer RksL
>
> **Q1: When switching from R50 to Swin-Large or even ViT-Large, does this approach of inserting a frozen CLIP still lead to noticeable improvement?**
>
> **RE**: Yes. We conduct experiments with Swin-L based on the Co-DETR detector. In this experiment, we equip the Co-DETR with DFN5B-CLIP-ViT-H-14-378 as the feature enhancer and fine-tune their Objects356 pre-trained checkpoint. Our Frozen-DETR is not used during the Objects365 pre-training. The results in the table below show that frozen-DETR can still gain noticeable improvement even with a strong backbone and strong pre-training.
>
> | Model                                        | AP   | AP$_{50}$ | AP$_{75}$ | AP$_{s}$ | AP$_m$ | AP$_l$ |
> |----------------------------------------------|------|-----------|-----------|----------|--------|--------|
> | Co-DINO 5scale SwinL (our re-implementation) | 63.7 | 81.2      | 70.2      | 50.2     | 67.1   | 77.8   |
> | Frozen-DETR                                  | 64.1 | 81.4      | 70.7      | 50.0     | 67.5   | 78.0   |
>
> ***
>
> **Q2: I am still curious whether the R50-CLIP would boost the performance of an R50 backbone detector. Does this increase necessitate that the model size of CLIP backbone surpasses that of the detector backbone?**
>
> **RE**: Here we use the same baseline as in Table 13. Using the R50-CLIP feature enhancer can still increase by 0.5% AP. Nevertheless, Frozen-DETR aims to equip detectors with a strong foundation model. If we use the R50 version of foundation models, we think using it as a backbone is a better choice.
>
> | Model               | AP   | AP$_{50}$ | AP$_{75}$ | AP$_{s}$ | AP$_m$ | AP$_l$ |
> |---------------------|------|-----------|-----------|----------|--------|--------|
> | baseline            | 47.0 | 64.1      | 51.4      | 30.5     | 50.2   | 62.0   |
> | baseline + R50-CLIP | 47.5 | 65.1      | 51.8      | 29.6     | 51.2   | 63.0   |

---

> > ### Comment · Reviewer_RksL · 2024-08-13
> > **Response to the authors**
> >
> > Thanks to the author for the response, the author's rebuttal has resolved my doubts. Regardless, I believe this article presents a simple but clever design.

---

### Official Review · Reviewer_V83J · 2024-07-15

**Soundness:** 3
**Presentation:** 3
**Contribution:** 3
**Rating:** 5
**Confidence:** 4

**Summary:**

This paper explores using frozen vision foundation models to enhance object detection performance without fine-tuning. The authors demonstrate that foundation models, although not pre-trained for object detection, can significantly improve detection accuracy by leveraging their high-level image understanding capabilities. This is achieved by using the class token from the foundation model to provide a global context and the patch tokens to enrich semantic details in the detection process. The proposed Frozen-DETR method boosts the state-of-the-art DINO detector's performance on the COCO dataset, achieving notable improvements in average precision (AP). The method also shows strong performance in large vocabulary settings and open-vocabulary detection, highlighting its robustness and versatility.

**Strengths:**

+ This paper proposes a relatively novel approach for leveraging visual foundation models for object detection.
+ The proposed method achieves promising results and outperforms other query-based object detection models.
+ The whole paper is easy to follow.

**Weaknesses:**

Overall I think this paper is interesting, their proposed method of using vision foundation models as additional knowledge is fair and relatively novel. One minor concern may be that, it is hard to say whether it is desirable to apply this relatively complicated framework instead of directly leveraging the pre-trained foundation models in real practice, but the improvement is solid based on the current evaluations.

**A few suggestions on writing:**

- I personally suggested splitting/reorganizing (e.g. one question + corresponding insight or one paragraph) the last paragraph of the introduction. The combination of questions and key insights is distractive.
- Figure 3 itself is hard to follow. Adding more details in the caption may help. (So as a few other figures/tables)
- It's better to bold the best results in tables.
- One minor comment: maybe use different notations for DINO (the detection baseline) and DINOv2 (the self-supervised encoder).



**Minor questions/comments:**

- l37-38, EVA-CLIP-18B is equipped with a ViT rather than R50. The version of CLIP using R50 as the backbone lagged behind. This evidence cannot support the claim that the improvement is from large-scale training.
- l55: Is DINO the SOTA query-based detector? (refer to https://paperswithcode.com/sota/object-detection-on-coco for example)
- The setup of Table 1 is a little unclear, is the backbone tunable? I assume they are not. Then, the comparison is slightly unfair -- the backbones are frozen for the top lines, and there are additional trainable parameters in the encoder backbone. In other words, if we finetune the encoder backbone of Table 1, will the gap between the two variants be even smaller, which questions the current design?
- In Table 2, DEiT-III has comparable results with CLIP, how will this backbone perform for the main experiments?
- One limitation of this approach is the running time. in Table 4, the authors reported the FPS, running time time has been 1.5x compared with the baseline model. However, if using the foundation model in a conventional way (i.e. directly using them as the backbone), the running time will not be affected in the inference time. Moreover, if using a ViT-based foundational model, how will the inference time and GPU memory change, since most of the recent powerful foundation models are equipped with ViT?

**Questions:**

See the previous section.

**Limitations:**

The authors discussed the limitations and the social impact at the end of the main paper. The limitation of this work may be that the visual foundation models are trained with natural images and they may not work well with images from other domains (e.g. medical images). There is no potential negative social impact from this work.

---

> ### Author Rebuttal · Authors · 2024-08-07
>
> # To Reviewer V83J
>
> **Q1: A few suggestions on writing**
>
> **RE**: Thanks for your helpful and detailed advice. We will carefully revise our manuscript. We rename the detector DINO as DINO-det and the self-supervised foundation model DINOv2 as DINOv2-FM.
>
> ***
>
> **Q2: EVA-CLIP-18B is equipped with a ViT rather than R50. The version of CLIP using R50 as the backbone lagged behind. This evidence cannot support the claim that the improvement is from large-scale training.**
>
> **RE**: This example aims to emphasize that a large-scale pre-trained foundation model has strong image understanding abilities such as zero-shot ability, even without fine-tuning on a specific downstream dataset. This property demonstrates that the foundation model has the potential to serve as a plug-and-play feature enhancer for downstream tasks. In the revision, we will replace the compared R50 with an ImageNet-22k pre-trained and ImageNet-1k fine-tuned ViT-H-14, which achieves 85.1% top1 accuracy and merely surpasses the zero-shot CLIP model by +1.3%, showing the foundation CLIP model's generalizable image understanding abilities.
>
> ***
>
> **Q3: Is DINO-det the SOTA query-based detector?**
>
> **RE**: DINO-det is one of the most well-known query-based detectors due to high performance and fast convergence speed and most recent SOTA detectors are based on it. Here we list some top detectors in the paperwithcode:
> - Co-DETR (the rank-1 detector) adds additional co-heads to DINO-det for fast and better convergence. We have also applied our method to Co-DETR in Table 6.
> - InternImage-H  (the rank-2 detector) and M3I Pre-training (the rank-3 detector) equip DINO-det with a strong backbone InternImage-H which is pre-trained by M3I Pre-training.
>
> ***
>
> **Q4: The setup of Table 1 is a little unclear, is the backbone tunable? I assume they are not. Then, the comparison is slightly unfair -- the backbones are frozen for the top lines, and there are additional trainable parameters in the encoder backbone. In other words, if we fine-tune the encoder backbone of Table 1, will the gap between the two variants be even smaller, which questions the current design?**
>
> **RE**: All backbones in Table 1 are tunable. Thus, the comparison is fair and the current design is reasonable. We will add more details to clarify the setup in Table 1.
>
> ***
>
> **Q5: In Table 2, DEiT-III has comparable results with CLIP, how will this backbone perform for the main experiments?**
>
> **RE**: We find DEiT-III can also boost the performance in the main experimental setup, but slightly lower than CLIP, as shown in the table below.
>
> | Model                      | AP   | AP$_{50}$ | AP$_{75}$ | AP$_{s}$ | AP$_m$ | AP$_l$ |
> |----------------------------|------|-----------|-----------|----------|--------|--------|
> | DINO-det-4scale            | 49.0 | 66.6      | 53.5      | 32.0     | 52.3   | 63.0   |
> | DINO-det-4scale + CLIP     | 51.9 | 70.4      | 56.7      | 33.8     | 54.9   | 69.3   |
> | DINO-det-4scale + DEiT-III | 50.6 | 68.5      | 55.0      | 32.1     | 53.5   | 67.9   |
>
> ***
>
> **Q6: One limitation of this approach is the running time. in Table 4, the authors reported the FPS, running time time has been 1.5x compared with the baseline model. However, if using the foundation model in a conventional way (i.e. directly using them as the backbone), the running time will not be affected in the inference time. Moreover, if using a ViT-based foundational model, how will the inference time and GPU memory change, since most of the recent powerful foundation models are equipped with ViT?**
>
> **RE**: We believe the additional computation cost is acceptable, which is also pointed out by Reviewer RksL. If we directly use ViT-L as the backbone, it will take 10G inference memory (3x larger than the baseline) and 2.1 FPS (4x slower than the baseline), which is much larger and slower than our Frozen-DETR. Please see the rebuttals for all reviewers above for more details.

---

> > ### Comment · Reviewer_V83J · 2024-08-13
> > **Thanks to the authors**
> >
> > Thanks to the authors for providing the detailed feedback. Most of my concerns are addressed and I am still leaerning a positive attitude towards this paper.

---

### Author Rebuttal · Authors · 2024-08-07

# To all reviewers

We thank all reviewers for their helpful and insightful feedback and are encouraged they find that our method is innovative (Reviewer V83J), the experiments are very comprehensive (Reviewer RksL and C9T7), and the proposed method achieves promising results (Reviewer V83J, RksL, and C9T7). We address reviewers' common comments below.

As mentioned by Reviewer V83J, the detector DINO and the self-supervised foundation model DINOv2 share the same name and may cause confusion. We rename the detector DINO as **DINO-det** and the self-supervised foundation model DINOv2 as **DINOv2-FM** in the following.

**Q1: Comparisons between using foundation models as a backbone and as a plug-and-play module as in our Frozen-DETR. Concerns for additional computation cost.**

| Method                           | Training | Training | Inference | Inference | GFLOPs |
|----------------------------------|:----------------:|:---------------:|:------------------:|:--------------:|:--------:|
|                                  | Mem                           | time / epoch                   | Mem    | FPS   |      |
|----------------------------------|----------------|---------------|------------------|--------------|--------|
| DINO-det-4scale baseline         | 13G                           | 1.3                            | 3G     | 9.7 | 279  |
|----------------------------------|----------------|---------------|------------------|--------------|--------|
| Our Frozen-DETR (DINO-det-4scale)    | 15G                           | 1.4                            | 3G     | 6.5 | 400  |
| DINO-det-5scale                  | 34G                           | 2.6                            | 5G     | 4.4 | 860  |
| DINO-det-4scale + ViT-L backbone | 44G (bs=1)                    | 4.2                            | 10G    | 2.1 | 1244 |


In this work, we propose a novel paradigm to integrate frozen vision foundation models with query-based detectors, firstly showing that frozen foundation models can be a versatile feature enhancer to boost the performance of detectors, **even though they are not pre-trained for object detection**. Please see the rebuttal PDF for intuitive comparisons among different paradigms.

In previous practices, large vision foundation models are always used as a pre-trained backbone and fine-tuned with detectors in an end-to-end manner. Although such a paradigm achieves a high performance, the computation cost of fine-tuning such a large vision foundation model is unaffordable. We use ViT-L as an example to illustrate this problem, as ViT-L is a common architecture for most vision foundation models. In the above table, we choose DINO-det-4scale with R50 backbone as the baseline and compare it with three methods: our Frozen-DETR (CLIP ViT-L-336), DINO-det-5scale, and DINO-det-4scale with a foundation model (ViT-L) as the backbone. We use the ViT-L as the backbone following ViTDet. For the training, we use 4 A100 GPUs with 2 images per GPU except for the ViT-L backbone due to out-of-memory (OOM). For inference, we use a V100 GPU with batch size 1 in line with the main text. As shown in the table, the computation cost in both training and inference for Frozen-DETR is the lowest among the three variants.
- Compared with DINO-det-4scale with a foundation model as a backbone, training a ViT-L backbone needs 4.2 hours per epoch and 44 GB memory per GPU, which is significantly higher than our Frozen-DETR (1.4 hours and 15 GB with 2 images per GPU). For inference, using ViT-L as a backbone needs 10 GB GPU memory and runs at 2.1 FPS on a V100 GPU. While inference with Frozen-DETR only needs 3 GB GPU memory (3x fewer) and runs at 6.5 FPS (3x faster).
- Compared with DINO-det-5scale, our Frozen-DETR not only runs faster but also significantly outperforms DINO-det-5scale by 1.8% AP (53.1% AP vs 51.3% AP), as shown in Table 6.

Thus, Frozen-DETR achieves a good performance-speed trade-off. The additional computation cost is acceptable and we are happy to find that **Reviewer RksL agrees with us**.

Apart from the good performance-speed trade-off, Frozen-DETR further enjoys the following advantages:
1. **No architecture constraint**. The foundation model in Frozen-DETR can be any architecture, including CNNs, ViTs, or hybrid ones. Moreover, the detector and the foundation model can use different structures. For example, the backbone of detectors can be R50 or Swin-B (Table 6) and the backbone of foundation models can be R101 and ViT (Table 13).
2. **Plug-and-play**. Our method can be plugged into various query-based detectors without modifying the detector’s structure, the foundation model’s structure, and the training recipe. We have applied Frozen-DETR to AdaMixer, DINO-det, Co-DETR in the main paper and DN-DETR, DAB-DETR, MS-DETR and HPR in the rebuttals below.
3. **Effective integration**. Our method can successfully transfer the strong image understanding ability from foundation models to detectors. We have shown that the benefit is larger under more challenging scenarios.
    - On the COCO dataset, we increase 2.9% AP for DINO-det (Table 6).
    - On the challenging large vocabulary LVIS dataset, we increase 6.6% AP for DINO-det (Table 7).
    - On the challenging long-tail scenario, we increase 8.7% APr and 7.7% APc for DINO-det (Table 7), showing the potential to alleviate the class imbalance problem.
    - On the challenging open-vocabulary scenario, we increase 8.8% novel AP for DINO-det (Table 8), showing strong open-vocabulary ability.
4. **Complementary to a strong backbone**. In Table 1 and Table 6, our Frozen-DETR can still boost the performance for detectors with CLIP initialized backbone or a strong ImageNet-22k pre-trained Swin-B.

Overall, we believe our novel paradigm enjoys good advantages over using foundation models as the backbone and achieves an outstanding performance-speed trade-off.

---

### Decision · Program_Chairs · 2024-09-25

**Decision:**

Accept (poster)

**Comment:**

This work studies how frozen foundation models can benefit object detection. After rebuttal, all the reviewers unanimously vote for acceptance of this work. Reviewers were concerned about the extra cost brought by the proposed method, and this issue has been thoroughly discussed in the rebuttal. After checking the rebuttal, the review, and the manuscript, the AC recommends acceptance.